# One Network Fits All? Modular versus Monolithic Task Formulations in Neural Networks

**Atish Agarwala & Abhimanyu Das**
Google Research
{thetish,abhidas}@google.com

**Brendan Juba**
Washington U. St. Louis*
bjuba@wustl.edu

**Rina Panigrahy**
Google Research
rinap@google.com

**Vatsal Sharan**
MIT†
vsharan@mit.edu

**Xin Wang & Qiuyi Zhang**
Google Research
{wanxin,qiuyiz}@google.com

## Abstract

Can deep learning solve multiple tasks simultaneously, even when they are unrelated and very different? We investigate how the representations of the underlying tasks affect the ability of a single neural network to learn them jointly. We present theoretical and empirical findings that a single neural network is capable of simultaneously learning multiple tasks from a combined data set, for a variety of methods for representing tasks—for example, when the distinct tasks are encoded by well-separated clusters or decision trees over certain task-code attributes. More concretely, we present a novel analysis that shows that families of simple programming-like constructs for the codes encoding the tasks are learnable by two-layer neural networks with standard training. We study more generally how the complexity of learning such combined tasks grows with the complexity of the task codes; we find that combining many tasks may incur a sample complexity penalty, even though the individual tasks are easy to learn. We provide empirical support for the usefulness of the learning bounds by training networks on clusters, decision trees, and SQL-style aggregation.

## 1 Introduction

Standard practice in machine learning has long been to only address carefully circumscribed, often very related tasks. For example, we might train a single classifier to label an image as containing objects from a certain predefined set, or to label the words of a sentence with their semantic roles. Indeed, when working with relatively simple classes of functions like linear classifiers, it would be unreasonable to expect to train a classifier that handles more than such a carefully scoped task (or related tasks in standard multitask learning). As techniques for learning with relatively rich classes such as neural networks have been developed, it is natural to ask whether or not such scoping of tasks is inherently necessary. Indeed, many recent works (see Section 1.2) have proposed eschewing this careful scoping of tasks, and instead training a single, "monolithic" function spanning many tasks.

Large, deep neural networks can, in principle, represent multiple classifiers in such a monolithic learned function (Hornik, 1991), giving rise to the field of multitask learning. This combined function might be learned by combining all of the training data for all of the tasks into one large batch–see Section 1.2 for some examples. Taken to an extreme, we could consider seeking to learn a *universal* circuit—that is, a circuit that interprets arbitrary programs in a programming language which can encode various tasks. But, the ability to *represent* such a monolithic combined function does not necessarily entail that such a function can be efficiently *learned* by existing methods. Cryptographic hardness theorems (Kearns & Valiant, 1994) establish that this is not possible in general by *any* method, let alone the specific training methods used in practice. Nevertheless, we still can ask how

---

*Work performed in part while visiting Google.
†Work performed in part while affiliated with Stanford, and in part while interning at Google.

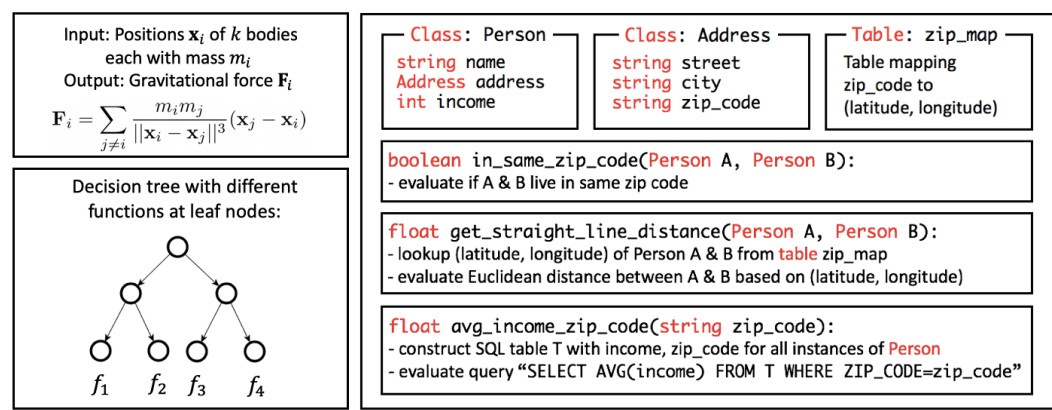

Figure 1: Our framework shows that it is possible to learn analytic functions such as the gravitational force law, decision trees with different functions at the leaf nodes, and programming constructs such as those on the right, all using a non-modular monolithic architecture.

rich a family of tasks can be learned by these standard methods. In this work, we study the extent to which backpropagation with stochastic gradient descent (SGD) can learn such monolithic functions on diverse, unrelated tasks. There might still be some inherent benefit to an architecture in which tasks are partitioned into sub-tasks of such small scope, and the training data is correspondingly partitioned prior to learning. For example, in the early work on multitask learning, Caruana (1997) observed that training a network to solve unrelated tasks simultaneously seemed to harm the overall performance. Similarly, the seminal work of Jacobs et al. (1991) begins by stating that *"If backpropagation is used to train a single, multilayer network to perform different subtasks on different occasions, there will generally be strong interference effects that lead to slow learning and poor generalization"*. We therefore ask if, for an unfortunate choice of tasks in our model, learning by standard methods might be fundamentally impaired.

As a point of reference from neuroscience, the classical view is that distinct tasks are handled in the brain by distinct patches of the cortex. While it is a subject of debate whether modularity exists for higher level tasks (Samuels, 2006), it is accepted that there are dedicated modules for low-level tasks such as vision and audio processing. Thus, it seems that the brain produces a *modular* architecture, in which different tasks are handled by different regions of the cortex. Conceivably, this division into task-specific regions might be driven by fundamental considerations of learnability: A single, monolithic neural circuit might simply be too difficult to learn because the different tasks might interfere with one another. Others have taken neural networks trained by backpropagation as a model of learning in the cortex (Musslick et al., 2017); to the extent that this is reasonable, our work has some bearing on these questions as well.

## 1.1 OUR RESULTS

We find, perhaps surprisingly, that combining multiple tasks into one cannot fundamentally impair learning with standard training methods. We demonstrate this for a broad family of methods for combining individual tasks into a single monolithic task. For example, inputs for each individual tasks may come from a disjoint region (for example, a disjoint ball) in a common input space, and each individual task could then involve applying some arbitrary simple function (e.g., a separate linear classifier for each region). Alternately there may be an explicit *"task code"* attribute (e.g., a one-hot code), together with the usual input attributes and output label(s), where examples with the same task code are examples for the same learning task. Complementing our results that combining multiple tasks does not impair learning, we also find that some task coding schemes do incur a sample complexity penalty.

A vast variety of task coding schemes may be used. As a concrete example, when the data points for each task are well-separated into distinct clusters, and the tasks are linear classification tasks, we show that a two-layer architecture trained with SGD successfully learns the combined, monolithic function; the required amount of data simply scales as the sum of the amount required to learn each

task individually (Theorem 2). Meanwhile, if the tasks are determined by a balanced decision tree of height $h$ on $d$ code attributes (as in Fig. 1, left), we find that the training time and amount of data needed scales as $\sim d^h$—quasipolynomial in the $2^h$ leaves (distinct tasks) when $d$ is of similar size to $h$, and thus when the coding is efficient (Theorem 3). We also prove a corresponding lower bound, which shows that this bound is in fact asymptotically tight (Theorem 3). More generally, for task codings based on decision trees using linear splits with a margin of at least $\gamma$ (when the data has unit $\ell_2$ norm), the training time and required data are asymptotically bounded by $\sim e^{O(h/\gamma^2)}$, which for constant $\gamma$ is polynomial in the $2^h$ functions (Theorem 4).

We generalize from these cluster-based and decision-tree based task codings to more complex codes that are actually simple programs. For instance, we show that SQL-style aggregation queries over a fixed database, written as a functions of the parameters of the query, can also be learned this way. More generally, simple programming constructs (such as in Fig. 1, right), built by operations such as compositions, aggregation, concatenation, and branching on a small number of such learnable functions, are also learnable (Theorem 5). In general, we can learn a low-depth formula (circuit with fan-out 1) in which each gate is not merely a switch (as in a decision tree), but can be any analytic function on the inputs, including arithmetic operations. Again, our key technical contribution is that we show that all of these functions are efficiently learned by SGD. This is non-trival since, although universal approximation theorems show that such functions can be expressed by (sufficiently wide) two-layer neural networks, under standard assumptions some expressible functions are not learnable Klivans & Sherstov (2009). We supplement the theoretical bounds with experiments on clusters, decision trees, and SQL-style aggregation showing that such functions are indeed learned in practice.

We note that the learning of such combined functions could have been engineered by hand: for example, there exist efficient algorithms for learning clusterings or such decision trees, and it is easy to learn the linear classifiers given the partitioned data. Likewise, these classes of functions are all known to be learnable by other methods, given an appropriate transformation of the input features. The key point is that *the two-layer neural network can jointly learn the task coding scheme and the task-specific functions without special engineering of the architecture*. That is, it is unnecessary to engineer a way of partitioning of the data into separate tasks prior to learning. Relatedly, the time and sample requirements of learning multiple tasks on a single network in general is insufficient to explain the modularity observed in biological neural networks if their learning dynamics are similar to SGD —i.e., we cannot explain the presence of modularity from such general considerations.

All our theoretical results are based upon a fundamental theorem that shows that analytic functions can be efficiently learnt by wide (but finite-width) two-layer neural networks with standard activation functions (such as ReLU), using SGD from a random initialization. Specifically, we derive novel generalization bounds for multivariate analytic functions (Theorems 1 and 8) by relating wide networks to kernel learning with a specific network-induced kernel (Jacot et al., 2018; Du et al., 2019; Allen-Zhu et al., 2019; Arora et al., 2019a; Lee et al., 2019), known as the *neural tangent kernel* (NTK) (Jacot et al., 2018). We further develop a *calculus of bounds* showing that the sum, product, ratio, and composition of analytic functions is also learnable, with bounds constructed using the familiar product and chain rules of univariate calculus (Corollaries 1, 2). These above learnability results may be of independent interest; for example, they can be used to show that natural physical laws like the gravitational force equations (shown in Fig. 1) can be efficiently learnt by neural networks (Section B.1). Furthermore, our bounds imply that the NTK kernel for ReLU activation has theoretical learning guarantees that are superior to the Gaussian kernel (Section A.2), which we also demonstrate empirically with experiments on learning the gravitational force law (Section B.2).

## 1.2 RELATED WORK

Most related to our work are a number of works in application areas that have sought to learn a single network that can perform many different tasks. In natural language processing, Tsai et al. (2019) show that a single model can solve machine translation across more than 50 languages. Many other works in NLP similarly seek to use one model for multiple languages, or even multiple tasks (Johnson et al., 2017; Aharoni et al., 2019; Bapna et al., 2019; Devlin et al., 2018). Monolithic models have also been successfully trained for tasks in very different domains, such as speech and language (Kaiser et al., 2017). Finally, there is also work on training extremely large neural networks which have the capacity to learn multiple tasks (Shazeer et al., 2017; Raffel et al., 2019). These works provide empirical clues

that suggest that a single network can successfully be trained to perform a wide variety of tasks. But, they do not provide a systematic theoretical investigation of the extent of this ability as we do here.

Caruana (1997) proposed *multitask learning* in which a single network is trained to solve multiple tasks on the same input simultaneously, as a vector of outputs. He observed that average generalization error for the multiple tasks may be much better than when the tasks are trained separately, and this observation initiated an active area of machine learning research (Zhang & Yang, 2017). Multitask learning is obviously related to our monolithic architectures. The difference is that whereas in multitask learning all of the tasks are computed simultaneously and output on separate gates, here all of the tasks share a common set of outputs, and the task code inputs switch between the various tasks. Furthermore, contrary to the main focus of multitask learning, we are primarily interested in the extent to which different tasks may interfere, rather than how much similar ones may benefit.

Our work is also related to studies of neural models of multitasking in cognitive science. In particular, Musslick et al. (2017) consider a similar two-layer architecture in which there is a set of task code attributes. But, as in multitask learning, they are interested in how many of these tasks can be performed simultaneously, on distinct outputs. They analyze the tradeoff between improved sample complexity and interference of the tasks with a handcrafted "gating" scheme, in which the parts of activity are zeroed out depending on the input (as opposed to the usual nonlinearities); in this model, they find out that the speedup from multitask learning comes at the penalty of limiting the number of tasks that can be correctly computed as the similarity of inputs varies. Thus, in contrast to our model where the single model is computing distinct tasks sequentially, they do find that the distinct tasks can interfere with each other when we seek to solve them simultaneously.

## 2  TECHNICAL OVERVIEW

We now give a more detailed overview of our theoretical techniques and results, with informal statements of our main theorems. For full formal statements and proofs, please see the Appendix.

### 2.1  LEARNING ANALYTIC FUNCTIONS

Our technical starting point is to generalize the analysis of Arora et al. (2019b) in order to show that two-layer neural networks with standard activation, trained by SGD from random initialization, can learn analytic functions on the unit sphere. We then obtain our results by demonstrating how our representations of interest can be captured by analytic functions with power series representations of appropriately bounded norms. Formal statements and proofs for this section appear in Appendix A.2. Let $S^d$ denote the unit sphere in $d$ dimensions.

**Theorem 1.** *(Informal) Given an analytic function $g(y)$, the function $g(\boldsymbol{\beta} \cdot \mathbf{x})$, for fixed $\boldsymbol{\beta} \in \mathbb{R}^d$ (with $\beta \overset{\text{def}}{=} \|\boldsymbol{\beta}\|_2$) and inputs $\mathbf{x} \in S^d$ is learnable to error $\epsilon$ with $n = O((\beta \tilde{g}'(\beta) + \tilde{g}(0))^2/\epsilon^2)$ examples using a single-hidden-layer, finite width neural network of width $\mathrm{poly}(n)$ trained with SGD, with*

$$\tilde{g}(y) = \sum_{k=0}^{\infty} |a_k| y^k \tag{1}$$

*where the $a_k$ are the power series coefficients of $g(y)$.*

We will refer to $\tilde{g}'(1)$ as the norm of the function $g$—this captures the Rademacher complexity of learning $g$, and hence the required sample complexity. We also show that the $\tilde{g}$ function in fact tightly captures the Rademacher complexity of learning $g$, i.e. there is a lower bound on the Rademacher complexity based on the coefficients of $\tilde{g}$ for certain input distributions (see Corollary 5 in Section C in the appendix).

We also note that we can prove a much more general version for multivariate analytic functions $g(\mathbf{x})$, with a modified norm function $\tilde{g}(y)$ constructed from the multivariate power series representation of $g(\mathbf{x})$ (Theorem 8 in Appendix A.2). The theorems can also be extended to develop a "calculus of bounds" which lets us compute new bounds for functions created via combinations of learnable functions. In particular, we have a product rule and a chain rule:

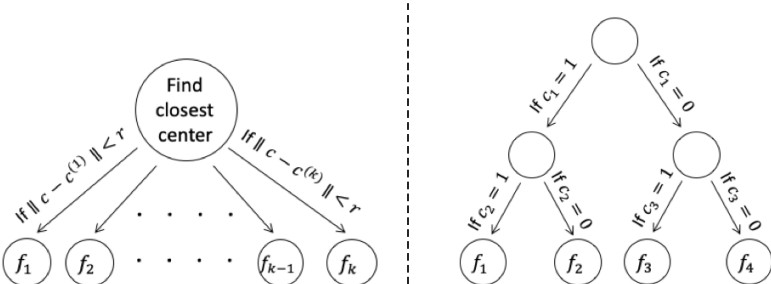

Figure 2: Some of the task codings which fit in our framework. On the left, we show a task coding via clusters. Here, $\mathbf{c}^{(i)}$ is the code for the $i$th cluster. On the right, we show a task coding based on low-depth decision trees. Here, $\mathbf{c}_i$ is the $i$th coordinate of the code $\mathbf{c}$ of the input datapoint.

**Corollary 1** (Product rule). *Let $g(\mathbf{x})$ and $h(\mathbf{x})$ meet the conditions of Theorem 1. Then the product $g(\mathbf{x})h(\mathbf{x})$ is efficiently learnable as well, with $O(M_{g \cdot h}/\epsilon^2)$ samples where*

$$\sqrt{M_{g \cdot h}} = \tilde{g}'(1)\tilde{h}(1) + \tilde{g}(1)\tilde{h}'(1) + \tilde{g}(0)\tilde{h}(0). \tag{2}$$

**Corollary 2** (Chain rule). *Let $g(y)$ be an analytic function and $h(\mathbf{x})$ be efficiently learnable, with auxiliary functions $\tilde{g}(y)$ and $\tilde{h}(y)$ respectively. Then the composition $g(h(\mathbf{x}))$ is efficiently learnable as well with $O(M_{g \circ h}/\epsilon^2)$ samples where*

$$\sqrt{M_{g \circ h}} = \tilde{g}'(\tilde{h}(1))\tilde{h}'(1) + \tilde{g}(\tilde{h}(0)), \tag{3}$$

*provided that $\tilde{h}(0)$ and $\tilde{h}(1)$ are in the radius of convergence of $g$.*

The calculus of bounds enables us to prove learning bounds on increasingly expressive functions, and we can prove results that may be of independent interest. As an example, we show in Appendix B.1 that forces on $k$ bodies interacting via Newtonian gravitation, as shown in Figure 1, can be learned to error $\epsilon$ using only $k^{O(\ln(k/\epsilon))}$ examples (even though the function $1/x$ has a singularity at 0).

## 2.2 TASK CODING VIA CLUSTERS

Our analysis of learning analytic functions allows us to prove that a single network with standard training can learn multiple tasks. We formalize the problem of learning multiple tasks as follows. In general, these networks take pairs of inputs $(\mathbf{c}, \mathbf{x})$ where $\mathbf{c}$ is a *task code* and $\mathbf{x}$ is the input (vector) for the chosen task represented by $\mathbf{c}$. We assume both $\mathbf{c}$ and $\mathbf{x}$ have fixed dimensionality. These pairs are then encoded by the concatenation of the two vectors, which we denote by $\mathbf{c}; \mathbf{x}$. Given $k$ tasks, corresponding to evaluation of functions $f_1, \ldots, f_k$ respectively on the input $\mathbf{x}$, the $i$th task has a corresponding code $\mathbf{c}^{(i)}$. Now, we wish to learn a function $g$ such that $g(\mathbf{c}^{(i)}; \mathbf{x}) = f_i(\mathbf{x})$ for examples of the form $(\mathbf{c}^{(i)}; \mathbf{x}, f_i(\mathbf{x}))$. This $g$ is a "monolithic" function combining the $k$ tasks. More generally, there may be some noise (bounded within a small ball around $\mathbf{c}^{(i)}$) in the task codes which would require learning the monolithic function $g(\mathbf{c}, x) = f_j(\mathbf{x})$ where $j = \operatorname{argmin}_i \|\mathbf{c} - \mathbf{c}^{(i)}\|_2$. Alternately the task-codes are not given explicitly but are inferred by checking which ball-center $\mathbf{c}^{(i)}$ (unique per task) is closest to the input $\mathbf{x}$ (see Fig. 2 (left) for an example). Note that these are all generalizations of a simple one-hot coding.

We assume throughout that the $f_i$ are analytic, with bounded-norm multinomial Taylor series representations. Our technical tool is the following Lemma (proved in Appendix A.2) which shows that the univariate step function $\mathbf{1}(x \geq 0)$ can be approximated with error $\epsilon$ and margin $\gamma$ using a low-degree polynomial which can be learnt using SGD.

**Lemma 1.** *Given a scalar $x$, let*

$$\Phi(x, \gamma, \epsilon) = (1/2)\left(1 + erf\left(Cx\sqrt{\log(1/\epsilon)}/\gamma\right)\right)$$

*where erf is the Gauss error function and $C$ is a constant. Let $\Phi'(x, \gamma, \epsilon)$ be the function $\Phi(x, \gamma, \epsilon)$ with its Taylor series truncated at degree $O(\log(1/\epsilon)/\gamma)$. Then,*

$$\Phi'(x, \gamma, \epsilon) = \begin{cases} O(\epsilon) & x \leq -\gamma/2, \\ 1 - O(\epsilon) & x \geq \gamma/2. \end{cases}$$

*Also, $\Phi'(x, \gamma, \epsilon)$ can be learnt using SGD with at most $e^{O((\log(1/\epsilon)/\gamma^2))}$ examples.*

Using this lemma, we show that indicator functions for detecting membership in a ball near a prototype $\mathbf{c}^{(i)}$ can also be sufficiently well approximated by functions with such a Taylor series representation. Specifically, we use the truncated representation of the erf function to indicate that $\|\mathbf{c} - \mathbf{c}^{(i)}\|$ is small. As long as the centers are sufficiently well-separated, we can find a low-degree, low-norm function this way using Lemma 1. For example, to check if $\mathbf{c}$ is within distance $r$ of center $\mathbf{c}^{(i)}$ we can use $\mathbf{1}(\|\mathbf{c} - \mathbf{c}^{(i)}\|^2 \leq r^2)$, which can be approximated using the $\phi'$ function in Lemma 1. Then given such approximate representations for the task indicators $I_1(\mathbf{c}), \ldots, I_k(\mathbf{c})$, the function $g(\mathbf{c}; \mathbf{x}) = I_1(\mathbf{c})f_1(\mathbf{x}) + \cdots + I_k(\mathbf{c})f_k(\mathbf{x})$ has norm linear in the complexities of the task functions, so that they are learnable by Theorem 1 (we scale to inputs to lie within the unit ball as required by Theorem 1). We state the result below, for the formal statement and proof see Appendix A.3.

**Theorem 2.** *(Informal) Given $k$ analytic functions having Taylor series representations with norm at most $poly(k/\epsilon)$ and degree at most $O(\log(k/\epsilon))$, a two-layer neural network trained with SGD can learn the following functions $g(\mathbf{c}; \mathbf{x})$ on the unit sphere to accuracy $\epsilon$ with sample complexity $poly(k/\epsilon)$ times the sum of the sample complexities for learning each of the individual functions:*

- *for $\Omega(1)$-separated codes $\mathbf{c}^{(1)}, \ldots, \mathbf{c}^{(k)}$, if $\|\mathbf{c} - \mathbf{c}^{(i)}\|_2 \leq O(1)$, then $g(\mathbf{c}; \mathbf{x}) = f_i(\mathbf{x})$.*

### 2.3 TASK CODING VIA LOW-DEPTH DECISION TREES

Theorem 2 can be viewed as performing a single $k$-way branching choice of which task function to evaluate. Alternatively, we can consider a sequence of such choices, and obtain a *decision tree* in which the leaves indicate which task function is to be applied to the input. We first consider the simple case of a decision tree when $\mathbf{c}$ is a $\{\pm 1\}$-valued vector. We can check that the values $c_1, \ldots, c_h$ match the fixed assignment $c_1^{(i)}, \ldots, c_h^{(i)}$ that reaches a given leaf of the tree using the function $I_{\mathbf{c}^{(i)}}(\mathbf{c}) = \prod_{j=1}^h \frac{c_j + c_j^{(i)}}{2}$ (or similarly for any subset of up to $h$ of the indices). Then $g(\mathbf{c}; \mathbf{x}) = I_{\mathbf{c}^{(1)}}(\mathbf{c})f_1(\mathbf{x}) + \cdots + I_{\mathbf{c}^{(k)}}(\mathbf{c})f_k(\mathbf{x})$ represents our decision tree coding of the tasks (see Fig. 2 (right) for an example). For the theorem, we again scale the inputs to lie within the unit ball:

**Theorem 3.** *(Informal) Two-layer neural networks trained with SGD can learn such a decision tree with depth $h$ within error $\epsilon$ with sample complexity $O(d^h/\epsilon^2)$ times the sum of the sample complexity for learning each of the individual functions at the leaves. Furthermore, conditioned on the hardness of learning parity with noise, $d^{\Omega(h)}$ examples are in fact necessary to learn a decision tree of depth $h$.*

We can generalize the previous decision tree to allow a threshold based decision at every internal node, instead of just looking at a coordinate. Assume that the input data lies in the unit ball and that each decision is based on a margin of at least $\gamma$. We can then use a product of our truncated erf polynomials to represent branches of the tree. We thus show:

**Theorem 4.** *(Informal) If we have a decision tree of depth $h$ where each decision is based on a margin of at least $\gamma$, then we can learn such a such a function within error $\epsilon$ with sample complexity $e^{O(h \log(1/\epsilon)/\gamma^2)}$ times the sample complexity of learning each of the leaf functions.*

For the formal statements and proofs, see Appendix A.4. Note that by Theorem 3, the exponential dependence on the depth in these theorems is necessary.

### 2.4 SIMPLE PROGRAMMING CONSTRUCTS

So far, we have discussed jointly learning $k$ functions with task codings represented by clusters and decision trees. We now move to a more general setup, where we allow simple programming constructs such as compositions, aggregation, concatenation, and branching on different functions. At this stage, the distinction between "task codes" and "inputs" becomes somewhat arbitrary. Therefore,

we will generally drop the task codes $\mathbf{c}$ from the inputs. The class of programming constructs we can learn is a generalization of the decision tree and we refer to it as a *generalized decision program*.

**Definition 1.** *We define a generalized decision program to be a circuit with fan-out 1 (i.e., a tree topology). Each gate in the circuit computes a function of the outputs of its children, and the root (top) node computes the final output. All gates, including the leaf gates, have access to the input $\mathbf{x}$.*

We can learn generalized decision programs where each node evaluates one among a large family of operations, first described informally below, and then followed by a formal definition.

**Arithmetic/analytic formulas**   As discussed in Section 2.1, learnability of analytic functions not only allows us to learn functions with bounded Taylor series, but also sums, products, and ratios of such functions. Thus, we can learn constant-depth arithmetic formulas with bounded outputs and analytic functions (with appropriately bounded Taylor series) applied to such learnable functions.

**Aggregation**   We observe that the sum of $k$ functions with bounded Taylor representations yields a function of the same degree and norm that is at most $k$ times greater; the average of these $k$ functions, meanwhile does not increase the magnitude of the norm. Thus, these standard aggregation operations are represented very efficiently. These enable us to learn functions that answer a family of SQL-style queries against a fixed database as follows: suppose $I(\mathbf{x}, r)$ is an indicator function for whether or not the record $r$ satisfies the predicate with parameters $\mathbf{x}$. Then a sum of the $m$ entries of a database that satisfy the predicate given by $\mathbf{x}$ is represented by $I(\mathbf{x}, r^{(1)})r^{(1)} + \cdots + I(\mathbf{x}, r^{(m)})r^{(m)}$. Thus, as long as the predicate function $I$ and records $r^{(i)}$ have bounded norms, the function mapping the parameters $\mathbf{x}$ to the result of the query is learnable. We remark that max aggregation can also be represented as a sum of appropriately scaled threshold indicators, provided that there is a sufficient gap between the maximum value and other values.

**Structured data**   We note that our networks already receive vectors of inputs and may produce vectors of outputs. Thus, one may trivially structured inputs and outputs such as those in Fig. 1 (right) using these vectors. We now formalize this by defining the class of functions we allow.

**Definition 2.** *We support the following operations at any gate in the generalized decision program. Let every gate have at most $k$ children. Let $g$ be the output of some gate and $\{f_1, \ldots, f_k\}$ be the outputs of the children of that gate.*

1. *Any analytic function of the child gates which can be approximated by a polynomial of degree at most $p$, including sum $g = \sum_{i=1}^{k} f_i$ and product of $p$ terms $g = \Pi_{i=1}^{p} f_i$.*
2. *Margin-based switch (decision) gate with children $\{f_1, f_2\}$ and some constant margin $\gamma$, i.e., $g = f_1$ if $\langle \boldsymbol{\beta}, \mathbf{x} \rangle - \alpha \leq -\gamma/2$, and $g = f_2$ if $\langle \boldsymbol{\beta}, \mathbf{x} \rangle - \alpha \geq \gamma/2$, for a vector $\boldsymbol{\beta}$ and constant $\alpha$.*
3. *Cluster-based switch gate with $k$ centers $\{\mathbf{c}^{(1)}, \ldots, \mathbf{c}^{(k)}\}$, with separation $r$ (for some constant $r$), i.e. the output is $f_i$ if $\|\mathbf{x} - \mathbf{c}^{(i)}\| \leq r/3$. A special case of this is a look-up table which returns value $v_i$ if $\mathbf{x} = \mathbf{c}^{(i)}$, and 0 if $\mathbf{x}$ does not match any of the centers.*
4. *Composition of two functions, $g(\mathbf{x}) = f_1(f_2(\mathbf{x}))$.*
5. *Create a tuple out of separate fields by concatenation: given inputs $\{f_1, \ldots, f_k\}$ $g$ outputs a tuple $[f_1, \ldots, f_k]$, which creates a single data structure out of the children. Or, extract a field out of a tuple: for a fixed field $i$, given the tuple $[f_1, \ldots, f_k]$, $g$ returns $f_i$.*
6. *For a fixed table $T$ with $k$ entries $\{r_1, \ldots, r_k\}$, a Boolean-valued function $b$, and an analytic function $f$, SQL queries of the form* `SELECT SUM f(r_i), WHERE b(r_i, x)` *for the input $\mathbf{x}$, i.e., $g$ computes $\sum_{i:b(r_i,\mathbf{x})=1} f(r_i)$. (We assume that $f$ takes bounded values and $b$ can be approximated by an analytic function of degree at most $p$.) For an example, see the function* `avg_income_zip_code()` *in Fig. 1 (right).*

As an example of a simple program we can support, refer to Fig. 1 (right) which involves table lookups, decision nodes, analytic functions such as Euclidean distance, and SQL queries. Theorem 5 is our learning guarantee for generalized decision programs. See Section A.5 in the Appendix for proofs, formal statements, and a detailed description of the program in Fig. 1 (right).

**Theorem 5.** *(Informal) Any generalized decision program of constant depth $h$ using the above operations with $p \leq O(\log(k/\epsilon))$ can be learnt within error $\epsilon$ with sample complexity $k^{poly(\log(k/\epsilon))}$. For the specific case of the program in Fig. 1 (right), it can be learnt using $(k/\epsilon)^{O(\log(1/\epsilon))}$ examples, where $k$ is the number of individuals in the database.*

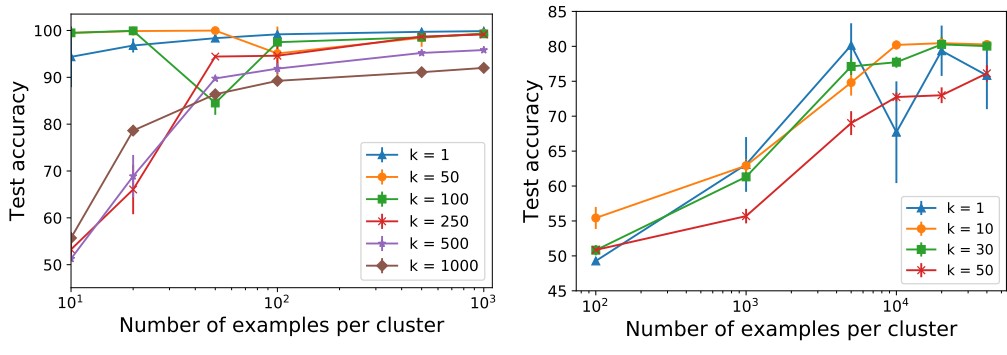

(a) Random linear classifier for each cluster.    (b) Random teacher network for each cluster.

Figure 3: Binary classification on multiple clusters, results are an average over 3 trials. A single neural network does well even when there are multiple clusters. The error does not increase substantially on increasing the number of clusters $k$

## 3    EXPERIMENTS

We next empirically explore the learnability of multiple functions by a two layer neural network when the tasks are coded by well-separated clusters or decision trees, and more generally the learnability of SQL-style aggregation for a fixed database. We find good agreement between the empirical performance and the bounds of Section 2. See Appendix D for more details of the experimental setup.

**Learning binary classification for well-separated clusters data**    We demonstrate through experiments on synthetic data that a single neural network can learn multiple tasks if the tasks are well-separated into clusters, as we discussed in Section 2.2. Here the data is drawn from a mixture of $k$ well-separated Gaussians in $d = 50$ dimensions. Within each Gaussian, the data points are marked with either of two labels. For the label generation, we consider two cases, first when the labels within each cluster are determined by a simple linear classifier, and second when the labels are given by a random teacher neural network with one hidden layer of 10 hidden units. Fig. 3 shows the performance of a single two-layer neural network with $50k$ hidden units on this task. The performance of the neural network changes only slightly on increasing the number of clusters $(k)$, suggesting that a single neural network can learn across all clusters.

**Learning polynomial functions on leaves of a decision tree**    We consider the problem of learning polynomial functions selected by a decision tree. The data generation process is as follows. We first fix parameters: tree depth $h$, decision variable threshold margin $\gamma$, number of variables $k$, and degree $p$ for leaf functions. Then we specify a full binary decision tree of depth $h$ with a random polynomial function on each leaf. To do this, we first generate thresholds $t_1, t_2, ..., t_h$ from the uniform distribution on $[0, 1]$ and $2^h$ leaf functions which are homogeneous polynomials of $k$ variables and degree $p$, with uniformly distributed random coefficients in $[0, 1]$. A train/test example $(\mathbf{x}, y)$ where $\mathbf{x} = (x_1, ..., x_h, x_{h+1}, ..., x_{h+p})$ is generated by first randomly sampling the $x_i$'s from the uniform distribution on $[0, 1]$, selecting the corresponding leaf based on $x_1, ..., x_h$ (that is, go left at the first branch if $x_1 \leq t_1$, otherwise go right, etc), and computing $y$ by evaluating the leaf function at $(x_{h+1}, ..., x_{h+p})$. The data is generated with the guarantee that each leaf has the same number of data points. Fig. 4 shows the performance of a two-layer neural network with $32 \times 2^h$ hidden units, measured in the R-squared metric. Here the R-squared metric is defined as $1 - \sum_i (\hat{y}_i - y_i)^2 / \sum_i (y_i - \overline{y})^2$, and is the fraction of the underlying variance explained by the model. Note that for a model that outputs the mean $\overline{y}$ for any input, the R-squared metric would be zero. We observed for a fixed number of training samples, accuracy increases as threshold margin increases, and the dependence of sample complexity on test error agrees with the bound in Theorem 4.

**Learning SQL-style aggregation queries**    We demonstrate the learnability of SQL-style aggregation queries,    which are functions of the form `SELECT SUM/MIN/MAX f(x)` `WHERE p(x)  from DATABASE`. The train and test datasets are generated from the Penn World

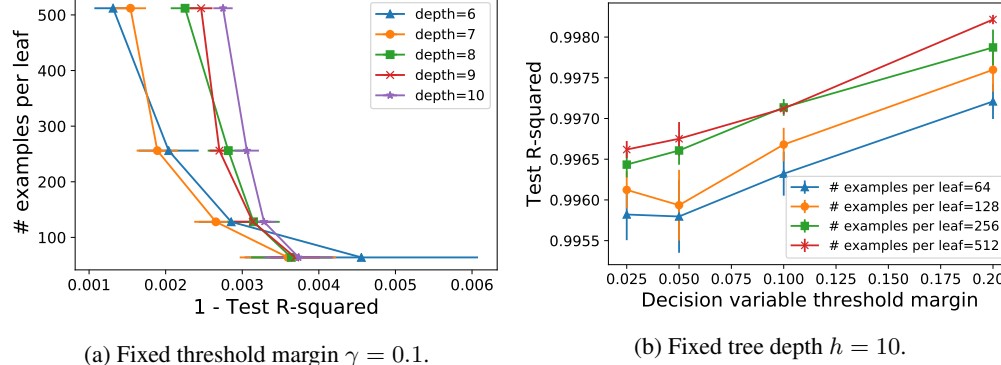

(a) Fixed threshold margin $\gamma = 0.1$.

(b) Fixed tree depth $h = 10$.

Figure 4: Learning random homogeneous polynomials of 4 variables and degree 4 on the leaves of a decision tree, the results are averaged over 7 trials. (a) Sample complexity scales as $e^{O(h \log(1/\epsilon)/\gamma^2)}$ with error $\epsilon$, where error is measured by (1-Test R-squared). (b) For fixed tree depth, accuracy increases with increasing margin.

Table dataset (Feenstra et al., 2015), which contains 11830 rows of economic data. The `WHERE` clause takes the form of $(x_{i_1} \geq t_{i_1})$ AND ... AND $(x_{i_k} \geq t_{i_k})$, where $x_{i_1}, \ldots, x_{i_k}$ are $k$ randomly selected columns and $t_{i_1}, \ldots, t_{i_k}$ are randomly selected values from the columns. The query target function is randomly selected from `SUM`, `MAX`, and `MIN` and is over a fixed column (`pl_x` in the table, which stands for price level for imports). The R-squared metric for a two-layer neural network with 40k hidden units is summarized in Table 1. We observe that a neural network learns to do SQL-style aggregation over dozens of data points, and for a fixed database, the test error only varies slightly for different numbers of columns in the `WHERE` clause.

Table 1: R-Squared for SQL-style aggregation. A single network with one hidden layer gets high R-Squared values, and the error does not increase substantially if the complexity of the aggregation is increased by increasing the number of columns in the WHERE clause.

| # columns in WHERE clause | 1 | 2 | 3 | 4 | 5 |
|---|---|---|---|---|---|
| Median # data points | 21 | 12 | 9 | 4 | 3 |
| Test R-Squared | $(93.31 \pm 0.11)$ % | $(93.01 \pm 2.7)$% | $(91.86 \pm 2.59)$ % | $(94.84 \pm 1.86)$ % | $(92.51 \pm 2.2)$ % |

## 4    CONCLUSION AND FUTURE WORK

Our results indicate that even using a single neural network, we can still learn tasks across multiple, diverse domains. However, modular architectures may still have benefits over monolithic ones: they might use less energy and computation, as only a portion of the total network needs to evaluate any data point. They may also be more interpretable, as it is clearer what role each part of the network is performing. It is an open question if any of these benefits of modularity can be extended to monolithic networks. For instance, is it necessary for a monolithic network to have modular parts which perform identifiable simple computations? And if so, can we efficiently identify these from the larger network? This could help in interpreting and understanding large neural networks.

Our work also begins to establish how neural networks can learn functions which are represented as simple programs. This perspective raises the question, how rich can these programs be? Can we learn programs from a full-featured language? In particular, supposing that they combine simpler programs using other basic operations such as composition, can such libraries of tasks be learned as well, i.e., can these learned programs be reused? We view this as a compelling direction for future work.

ACKNOWLEDGEMENTS

Brendan Juba was partially supported by NSF Awards CCF-1718380, IIS-1908287, and IIS-1939677, and was visiting Google during a portion of this work. Vatsal Sharan was supported in part by NSF award 1704417.

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

# A    THEORETICAL RESULTS

## A.1    KERNEL LEARNING BOUNDS

In this section, we develop the theory of learning analytic functions. For a given function $g$, we define a parameter $M_g$ related to the sample complexity of learning $g$ with small error with respect to a given loss function:

**Definition 3.** *Fix a learning algorithm, and a 1-Lipschitz loss function $\mathcal{L}$. For a function $g$ over a distribution of inputs $\mathcal{D}$, a given error scale $\epsilon$, and a confidence parameter $\delta$, let the* sample complexity $n_{g,\mathcal{D}}(\epsilon, \delta)$ *be the smallest integer such that when the algorithm is given $n_{g,\mathcal{D}}(\epsilon, \delta)$ i.i.d. examples of $g$ on $\mathcal{D}$, with probability greater than $1 - \delta$, it produces a trained model $\hat{g}$ with generalization error $\mathrm{E}_{\mathbf{x} \sim \mathcal{D}}[\mathcal{L}(g(\mathbf{x}), \hat{g}(\mathbf{x}))]$ less than $\epsilon$. Fix a constant $C > 0$. We say $g$ is* efficiently learned *by the algorithm (w.r.t. $C$) if there exists a constant $M_g$ (depending on $g$) such that for all $\epsilon$, $\delta$, and distributions $\mathcal{D}$ on the inputs of $g$, $n_{g,\mathcal{D}}(\epsilon, \delta) \leq C([M_g + \log(\delta^{-1})]/\epsilon^2)$.*

For example, it is known (Talagrand (1994)) that there exists a suitable choice of $C$ such that empirical risk minimization for a class of functions efficiently learns those functions with $M_g$ at most the VC-dimension of that class.

Previous work focused on computing $M_g$, for functions defined on the unit sphere, for wide neural networks trained with SGD. We extend the bounds derived in Arora et al. (2019a) to analytic functions, and show that they apply to kernel learning methods as well as neural networks.

The analysis in Arora et al. (2019a) focused on the case of training the hidden layers of wide networks with SGD. We first show that these bounds are more general and in particular apply to the case where only the final layer weights are trained (corresponding to the NNGP kernel in Lee et al. (2019)), and therefore our results will apply to general kernel learning as well. The proof strategy consists of showing that finite-width networks have a sensible infinite-width limit, and showing that training causes only a small change in parameters of the network.

Let $m$ be the number of hidden units, and $n$ be the number of data points. Let $\mathbf{y}$ be the $n \times 1$ dimensional vector of training outputs. Let $\mathbf{h}$ be a $n \times m$ random matrix denoting the activations of the hidden layer (as a function of the weights of the lower layer) for all $n$ data points. We will first show the following:

**Theorem 6.** *For sufficiently large $m$, a function $g$ can be learned efficiently in the sense of Definition 3 by training the final layer weights only with SGD, where the constant $M_g$ given by*

$$M_g \leq \mathbf{y}^{\mathrm{T}}(\mathbf{H}^\infty)^{-1}\mathbf{y} \tag{4}$$

*where we define $\mathbf{H}^\infty$ as*

$$\mathbf{H}^\infty = \mathrm{E}[\mathbf{h}\mathbf{h}^{\mathrm{T}}] \tag{5}$$

*which is the NNGP kernel from Lee et al. (2019).*

We require some technical lemmas in order to prove the theorem. We first need to show that $\mathbf{H}^\infty$ is, with high probability, invertible. If $K(\mathbf{x}, \mathbf{x}')$, the kernel function which generates $\mathbf{H}^\infty$ is given by a infinite Taylor series in $\mathbf{x} \cdot \mathbf{x}'$ it can be argued that $\mathbf{H}^\infty$ has full rank for most real world distributions. For example, the ReLU activation this holds as long as no two data points are co-linear (see Definition 5.1 in Arora et al. (2019a)). We can prove this more explicitly in the following lemma:

**Lemma 2.** *If all the $n$ data points $x$ are distinct and the Taylor series of $K(\mathbf{x}, \mathbf{x}')$ in $\mathbf{x} \cdot \mathbf{x}'$ has positive coefficients everywhere then $\mathbf{H}^\infty$ is not singular.*

*Proof.* First consider the case where the input $x$ is a scalar. Since the Taylor series corresponding to $K(x, x')$ consists of monomials of all degrees of $xx'$, we can view it as some inner product in a kernel space induced by the function $\phi(x) = (1, x, x^2, \ldots)$, where the inner product is diagonal (but with potentially different weights) in this basis. For any distinct set of inputs $\{x_1, .., x_n\}$ the set of vectors $\phi(x_i)$ are linearly independent. The first $n$ columns produce the Vandermonde matrix obtained by stacking rows $1, x, x, ..., x^{n-1}$ for $n$ different values of $x$, which is well known to be non-singular (since a zero eigenvector would correspond to a degree $n - 1$ polynomial with $n$ distinct roots $\{x_1, .., x_n\}$).

This extends to the case of multidimensional $\mathbf{x}$ if the values, projected along some dimension, are distinct. In this case, the kernel space corresponds to the direct sum of copies of $\phi$ applied elementwise to each coordinate $\mathbf{x}_i$. If all the points are distinct and and far apart from each other, the probability that a given pair coincides under random projection is negligible. From a union bound, the probability that a given pair coincide is also bounded – so there must be directions such that projections along that direction are distinct. Therefore, $\mathbf{H}^\infty$ can be considered to be invertible in general. $\qquad\square$

As $m \to \infty$, $\mathbf{h}\mathbf{h}^{\mathrm{T}}$ concentrates to its expected value. More precisely, $(\mathbf{h}\mathbf{h}^{\mathrm{T}})^{-1}$ approaches $(\mathbf{H}^\infty)^{-1}$ for large $m$ if we assume that the smallest eigenvalue $\lambda_{min}(\mathbf{H}^\infty) \geq \lambda_0$, which from the above lemma we know to be true for fixed $n$. (For the ReLU NTK the difference becomes negligible with high probability for $m = poly(n/\lambda_0)$ Arora et al. (2019a).) This allows us to replace $\mathbf{h}\mathbf{h}^{\mathrm{T}}$ with $\mathbf{H}^\infty$ in any bounds involving the former.

We can get learning bounds in terms of $\mathbf{h}\mathbf{h}^{\mathrm{T}}$ by studying the upper layer weights $\mathbf{w}$ of the network after training. After training, we have $\mathbf{y} = \mathbf{w} \cdot \mathbf{h}$. If $\mathbf{h}\mathbf{h}^{\mathrm{T}}$ is invertible (which the above arguments show is true with high probability for large $m$), the following lemma holds:

**Lemma 3.** *If we initialize a random lower layer and train the weights of the upper layer, then there exists a solution $\mathbf{w}$ with norm $\sqrt{\mathbf{y}^{\mathrm{T}}(\mathbf{h}\mathbf{h}^{\mathrm{T}})^{-1}\mathbf{y}}$.*

*Proof.* The minimum norm solution to $\mathbf{y} = \mathbf{w}^{\mathrm{T}}\mathbf{h}$ is

$$\mathbf{w}^* = (\mathbf{h}^{\mathrm{T}}\mathbf{h})^{-1}\mathbf{h}^{\mathrm{T}}\mathbf{y}. \tag{6}$$

The norm squared $(\mathbf{w}^*)^{\mathrm{T}}\mathbf{w}^*$ of this solution is given by $\mathbf{y}^{\mathrm{T}}\mathbf{h}(\mathbf{h}^{\mathrm{T}}\mathbf{h})^{-2}\mathbf{h}^{\mathrm{T}}\mathbf{y}$.

We claim that $\mathbf{h}(\mathbf{h}^{\mathrm{T}}\mathbf{h})^{-2}\mathbf{h}^{\mathrm{T}} = (\mathbf{h}\mathbf{h}^{\mathrm{T}})^{-1}$. To show this, consider the SVD decomposition $\mathbf{h} = \mathbf{U}\mathbf{S}\mathbf{V}^{\mathrm{T}}$. Expanding we have

$$\mathbf{h}(\mathbf{h}^{\mathrm{T}}\mathbf{h})^{-2}\mathbf{h}^{\mathrm{T}} = \mathbf{U}\mathbf{S}\mathbf{V}^{\mathrm{T}}(\mathbf{V}\mathbf{S}^2\mathbf{V}^{\mathrm{T}})^{-2}\mathbf{V}\mathbf{S}\mathbf{U}^{\mathrm{T}}. \tag{7}$$

Evaluating the right hand side gets us $\mathbf{U}\mathbf{S}^{-2}\mathbf{U}^{\mathrm{T}} = (\mathbf{h}\mathbf{h}^{\mathrm{T}})^{-1}$.

Therefore, the norm of the minimum norm solution is $\mathbf{y}^{\mathrm{T}}(\mathbf{h}\mathbf{h}^{\mathrm{T}})^{-1}\mathbf{y}$. $\qquad\square$

We can now complete the proof of Theorem 6.

*Proof of Theorem 6.* For large $m$, the squared norm of the weights approaches $\mathbf{y}^{\mathrm{T}}(\mathbf{H}^\infty)^{-1}\mathbf{y}$. Since the lower layer is fixed, the optimization problem is linear and therefore convex in the trained weights $\mathbf{w}$. Therefore SGD with small learning rate will reach this optimal solution. The Rademacher complexity of this function class is at most $\sqrt{\mathbf{y}^{\mathrm{T}}(\mathbf{H}^\infty)^{-1}\mathbf{y}}$ which we at most by $\sqrt{M_g}$ where $M_g$ is an upper bound on $\mathbf{y}^{\mathrm{T}}(\mathbf{H}^\infty)^{-1}\mathbf{y}$. The optimal solution has $0$ train error based on the assumption that $\mathbf{H}^\infty$ is full rank and the generalization error will be no more than $O(\sqrt{\frac{\mathbf{y}^{\mathrm{T}}(\mathbf{H}^\infty)^{-1}\mathbf{y}}{2n}})$ which is at most $\epsilon$ if we use at least $n = \Omega(M_g/\epsilon^2)$ training samples - note that this is identical to the previous results for training the hidden layer only Arora et al. (2019a); Du et al. (2019). $\qquad\square$

### A.2 LEARNING ANALYTIC FUNCTIONS

Now, we derive our generalization bounds for single variate functions. We use Theorem 6 to prove the following corollary, a more general version of Corollary 6.2 proven in Arora et al. (2019a) for wide ReLU networks with trainable hidden layer only:

**Corollary 3.** *Consider the function $g : \mathbb{R}^d \to \mathbb{R}$ given by:*

$$g(\mathbf{x}) = \sum_k a_k(\boldsymbol{\beta}_k^{\mathrm{T}}\mathbf{x})^k \tag{8}$$

*Then, if $g$ is restricted to $||\mathbf{x}|| = 1$, and the NTK or NNGP kernel can be written as $H(\mathbf{x}, \mathbf{x}') = \sum_k b_k(\mathbf{x} \cdot \mathbf{x}')^k$, the function can be learned efficiently with a wide one-hidden-layer network in the sense of Definition 3 with*

$$\sqrt{M_g} = \sum_k b_k^{-1/2}|a_k|||\boldsymbol{\beta}_k||_2^k \tag{9}$$

up to g-independent constants of $O(1)$, where $\beta_k \equiv ||\boldsymbol{\beta}_k||_2$. *In the particular case of a ReLU network, the bound is*

$$\sqrt{M_g} = \sum_k k|a_k|||\boldsymbol{\beta}_k||_2^k \tag{10}$$

The original corollary applied only to networks with trained hidden layer, and the bound on the ReLu network excluded odd monomials of power greater than 1.

*Proof.* The extension to NNGP follows from Theorem 6, which allows for the application of the arguments used to prove Corollary 6.2 from Arora et al. (2019a) (particularly those found in Appendix E).

The extension of the ReLu bound to odd powers can be acheived with the following modification. consider appending a constant component to the input $\mathbf{x}$ so that the new input to the network is $(\mathbf{x}/\sqrt{2}, 1/\sqrt{2})$. The kernel then becomes:

$$K(\mathbf{x}, \mathbf{x}') = \frac{\mathbf{x} \cdot \mathbf{x}' + 1}{4\pi}\left(\pi - \arccos\left(\frac{\mathbf{x} \cdot \mathbf{x}' + 1}{2}\right)\right). \tag{11}$$

Re-writing the power series as an expansion around $\mathbf{x} \cdot \mathbf{x}' = 0$, we have terms of all powers. An asymptotic analysis of the coefficients using known results shows that coefficients $b_k$ are asymptotically $O(k^{-3/2})$ - meaning in Equation 10 applies to these kernels, without restriction to even $k$. $\square$

Equation 9 suggests that kernels with slowly decaying (but still convergent) $b_k$ will give the best bounds for learning polynomials. Many popular kernels do not meet this criteria. For example, for inputs on the sphere of radius $r$, the Gaussian kernel $K(\mathbf{x}, \mathbf{x}') = e^{-||\mathbf{x}-\mathbf{x}'||^2/2}$ can be written as $K(\mathbf{x}, \mathbf{x}') = e^{-r^2}e^{\mathbf{x} \cdot \mathbf{x}'}$. This has $b_k^{-1/2} = e^{r^2/2}\sqrt{k!}$, which increases rapidly with $k$. This provides theoretical justification for the empirically inferior performance of the Gaussian kernel which we will present in Section B.2.

Guided by this theory, we focus on kernels where $b_k^{-1/2} \leq O(k)$, for all $k$ (or, $b_k \geq O(k^{-2})$). The modified ReLu meets this criterion, as well as hand-crafted kernels of the form

$$K(\mathbf{x}, \mathbf{x}') = \sum_k k^{-s}(\mathbf{x} \cdot \mathbf{x}')^k \tag{12}$$

with $s \in (1, 2]$ is a valid slowly decaying kernel on the sphere. We call these slowly decaying kernels. We note that by Lemma 3, the results of Corollary 3 apply to networks with output layer training only, as well as kernel learning (which can be implemented by training wide networks).

Using the extension of Corollary 3 to odd powers, we first show that analytic functions with appropriately bounded norms can be learnt.

**Theorem 7.** *Let $g(y)$ be a function analytic around $0$, with radius of convergence $R_g$. Define the auxiliary function $\tilde{g}(y)$ by the power series*

$$\tilde{g}(y) = \sum_{k=0}^{\infty} |a_k|y^k \tag{13}$$

*where the $a_k$ are the power series coefficients of $g(y)$. Then the function $g(\boldsymbol{\beta} \cdot \mathbf{x})$, for some fixed vector $\boldsymbol{\beta} \in \mathbb{R}^d$ with $||\mathbf{x}|| = 1$ is efficiently learnable in the sense of Definition 3 using a model with a slowly decaying kernel $K$ with*

$$\sqrt{M_g} = \beta\tilde{g}'(\beta) + \tilde{g}(0) \tag{14}$$

*if the norm $\beta \equiv ||\boldsymbol{\beta}||_2$ is less than $R_g$.*

*Proof.* We first note that the radius of convergence of the power series of $\tilde{g}(y)$ is also $R_g$ since $g(y)$ is analytic. Applying Equation 10, pulling out the 0th order term, and factoring out $\beta$, we get

$$\sqrt{M_g} = |a_0| + \beta\sum_{k=1}^{\infty} k|a_k|\beta^k = \beta\tilde{g}'(\beta) + \tilde{g}(0) \tag{15}$$

since $\beta < R_g$. $\square$

The tilde function is the notion of complexity which measures how many samples we need to learn a given function. Informally, the tilde function makes all coefficients in the Taylor series positive. The sample complexity is given by the value of the function at 1 (in other words, the L1 norm of the coefficients in the Taylor series). For a multivariate function $g(\mathbf{x})$, we define its tilde function $\tilde{g}(y)$ by substituting any inner product term $\langle \boldsymbol{\alpha}, \mathbf{x} \rangle$ by a univariate $y$. The above theorem can then also be generalized to multivariate analytic functions:

**Lemma 4.** *Given a collection of $p$ vectors $\boldsymbol{\beta}_i$ in $\mathbb{R}^d$, the function $f(\mathbf{x}) = \prod_{i=1}^p \boldsymbol{\beta}_i \cdot \mathbf{x}$ is efficiently learnable with*

$$\sqrt{M_f} = p \prod_i \beta_i \tag{16}$$

*where $\beta_i \equiv ||\boldsymbol{\beta}_i||_2$.*

*Proof.* The proof of Corollary 6.2 in Arora et al. (2019a) relied on the following statement: given positive semi-definite matrices $\mathbf{A}$ and $\mathbf{B}$, with $\mathbf{A} \succeq \mathbf{B}$, we have:

$$\mathbf{P_B A}^{-1} \mathbf{P_B} \preceq \mathbf{B}^+ \tag{17}$$

where $+$ is the Moore-Penrose pseudoinverse, and $\mathbf{P}$ is the projection operator.

We can use this result, along with the Taylor expansion of the kernel and a particular decomposition of a multivariate monomial in the following way. Let the matrix $\mathbf{X}$ to be the training data, such that the $\alpha$th column $\mathbf{x}_i$ is a unit vector in $\mathbb{R}^d$. Given $\mathbf{K} \equiv \mathbf{X}^{\mathrm{T}} \mathbf{X}$, the matrix of inner products, the Gram matrix $\mathbf{H}^\infty$ of the kernel can be written as

$$\mathbf{H}^\infty = \sum_{k=0}^\infty b_k \mathbf{K}^{\circ k} \tag{18}$$

where $\circ$ is the Hadamard (elementwise) product. Consider the problem of learning the function $f(\mathbf{x}) = \prod_{i=1}^p \boldsymbol{\beta}_i \cdot \mathbf{x}$. Note that we can write:

$$f(\mathbf{X}) = (\mathbf{X}^{\circ k})^{\mathrm{T}} \otimes_{i=1}^k \boldsymbol{\beta}_i. \tag{19}$$

Here $\otimes$ is the tensor product, which for vectors takes an $n_1$-dimensional vector and an $n_2$ dimensional vector as inputs vectors and returns a $n_1 n_2$ dimensional vector:

$$\mathbf{w} \otimes \mathbf{v} = \begin{pmatrix} w_1 v_1 \\ w_1 v_2 \\ \cdots \\ w_1 v_{n_2} \\ w_2 v_1 \\ \cdots \\ w_{n_1} v_{n_2} \end{pmatrix}. \tag{20}$$

The operator $\odot$ is the Khatri-Rao product, which takes an $n_1 \times n_3$ matrix $\mathbf{A} = (\mathbf{a}_1, \cdots, \mathbf{a}_{n_3})$ and a $n_2 \otimes n_3$ matrix $\mathbf{B} = (\mathbf{b}_1, \cdots, \mathbf{b}_{n_3})$ and returns the $n_1 n_2 \times n_3$ dimensional matrix

$$\mathbf{A} \odot \mathbf{B} = (\mathbf{a}_1 \otimes \mathbf{b}_1, \cdots, \mathbf{a}_{n_3} \otimes \mathbf{b}_{n_3}). \tag{21}$$

For $p = 2$, this form of $f(\mathbf{X})$ can be proved explicitly:

$$(\mathbf{X}^{\circ 2})^{\mathrm{T}} \boldsymbol{\beta}_1 \otimes \boldsymbol{\beta}_2 = (\mathbf{x}_1 \otimes \mathbf{x}_1, \cdots, \mathbf{x}_P \otimes \mathbf{x}_P)^{\mathrm{T}} \boldsymbol{\beta}_1 \otimes \boldsymbol{\beta}_2. \tag{22}$$

The $\alpha$th element of the matrix product is

$$(\mathbf{x}_\alpha \otimes \mathbf{x}_\alpha) \cdot (\boldsymbol{\beta}_1 \otimes \boldsymbol{\beta}_2) = (\boldsymbol{\beta}_1 \cdot \mathbf{x}_\alpha)(\boldsymbol{\beta}_2 \cdot \mathbf{x}_\alpha) \tag{23}$$

which is exactly $f(\mathbf{x}_\alpha)$. The formula can be proved for $p > 2$ by finite induction.

With this form of $f(\mathbf{X})$, we can follow the steps of the proof in Appendix E of Arora et al. (2019a), which was written for the case where the $\boldsymbol{\beta}_i$ were identical:

$$\mathbf{y}^{\mathrm{T}} (\mathbf{H}^\infty)^{-1} \mathbf{y} = (\otimes_{i=1}^p \boldsymbol{\beta}_i)^{\mathrm{T}} \mathbf{X}^{\circ p} (\mathbf{H}^\infty)^{-1} (\mathbf{X}^{\circ p})^{\mathrm{T}} \otimes_{i=1}^p \boldsymbol{\beta}_i. \tag{24}$$

Using Equation 17, applied to $\mathbf{K}^{\circ p}$, we have:

$$\mathbf{y}^{\mathrm{T}}(\mathbf{H}^\infty)^{-1}\mathbf{y} \leq$$
$$b_p^{-1}(\otimes_{i=1}^p \boldsymbol{\beta}_i)^{\mathrm{T}}\mathbf{X}^{\odot p}\mathbf{P}_{\mathbf{K}^{\circ p}}(\mathbf{K}^{\circ p})^+\mathbf{P}_{\mathbf{K}^{\circ p}}(\mathbf{X}^{\odot p})^{\mathrm{T}}\otimes_{i=1}^p\boldsymbol{\beta}_i . \tag{25}$$

Since the $\mathbf{X}^{\odot p}$ are eigenvectors of $\mathbf{P}_{\mathbf{K}^{\circ p}}$ with eigenvalue 1, and $\mathbf{X}^{\odot p}(\mathbf{K}^{\circ p})^+(\mathbf{X}^{\odot p})^{\mathrm{T}} = \mathbf{P}_{\mathbf{X}^{\odot p}}$, we have:

$$\mathbf{y}^{\mathrm{T}}(\mathbf{H}^\infty)^{-1}\mathbf{y} \leq b_p^{-1}(\otimes_{i=1}^p\boldsymbol{\beta}_i)^{\mathrm{T}}\mathbf{P}_{\mathbf{X}^{\odot p}}\otimes_{i=1}^p\boldsymbol{\beta}_i \tag{26}$$

$$\mathbf{y}^{\mathrm{T}}(\mathbf{H}^\infty)^{-1}\mathbf{y} \leq b_p^{-1}\prod_{i=1}^p\boldsymbol{\beta}_i\cdot\boldsymbol{\beta}_i . \tag{27}$$

For the slowly decaying kernels, $b_p \geq p^{-2}$. Therefore, we have $\sqrt{\mathbf{y}^{\mathrm{T}}(\mathbf{H}^\infty)^{-1}\mathbf{y}} \leq \sqrt{M_f}$ for

$$\sqrt{M_f} = p\prod_i\beta_i \tag{28}$$

where $\beta_i \equiv ||\boldsymbol{\beta}_i||_2$, as desired. $\qquad\square$

This leads to the following generalization of Theorem 7:

**Theorem 8.** *Let $g(\mathbf{x})$ be a function with multivariate power series representation:*

$$g(\mathbf{x}) = \sum_k\sum_{v\in V_k}a_v\prod_{i=1}^k(\boldsymbol{\beta}_{v,i}\cdot\mathbf{x}) \tag{29}$$

*where the elements of $V_k$ index the $k$th order terms of the power series. We define $\tilde{g}(y) = \sum_k \tilde{a}_k y^k$ with coefficients*

$$\tilde{a}_k = \sum_{v\in V_k}|a_v|\prod_{i=1}^k\beta_{v,i}. \tag{30}$$

*If the power series of $\tilde{g}(y)$ converges at $y = 1$ then with high probability $g(\mathbf{x})$ can be learned efficiently in the sense of Definition 3 with $\sqrt{M_g} = \tilde{g}'(1) + \tilde{g}(0)$.*

*Proof.* Follow the construction in Theorem 7, using Lemma 4 to get bounds on the individual terms. Then sum and evaluate the power series of $\tilde{g}'(1)$ to arrive at the bound. $\qquad\square$

**Remark 1.** *Note that the $\tilde{g}$ function defined above for multivariate functions depends on the representation, i.e. choice of the vectors $\boldsymbol{\beta}$. Therefore to be fully formal $\tilde{g}(y)$ should instead be $\tilde{g}_{\boldsymbol{\beta}}(y)$. For clarity, we drop $\boldsymbol{\beta}$ from the expression $\tilde{g}_{\boldsymbol{\beta}}(y)$ and it is implicit in the $\tilde{g}$ notation.*

**Remark 2.** *If $g(\mathbf{x})$ can be approximated by some function $g_{app}$ such that $|g(\mathbf{x}) - g_{app}| \leq \epsilon'$ for all $\mathbf{x}$ in the unit ball, then Theorem 8 can be used to learn $g(\mathbf{x})$ within error $\epsilon' + \epsilon$ with sample complexity $O(M_{g_{app}}/\epsilon^2)$.*

To verify Remark 2, note that we are doing regression on the upper layer of the neural network, where the lower layer is random. So based on $g_{\mathrm{app}}$ there exists a low-norm solution for the regression coefficients for the upper layer weights which gets error at most $\epsilon'$. If we solve the regression under the appropriate norm ball, then we get training error at most $\epsilon'$, and the generalization error will be at most $\epsilon$ with $O(M_{g_{\mathrm{app}}}/\epsilon^2)$ samples.

We can also derive the equivalent of the product and chain rule for function composition.

*Proof of Corollary 1.* Consider the power series of $g(\mathbf{x})h(\mathbf{x})$, which exists and is convergent since each individual series exists and is convergent. Let the elements of $V_{j,g}$ and $V_{k,h}$ index the $j$th order terms of $g$ and the $k$th order terms of $h$ respectively. The individual terms in the series look like:

$$a_v b_w\prod_{j'=1}^j(\boldsymbol{\beta}_{v,j'}\cdot\mathbf{x})\prod_{k'=1}^k(\boldsymbol{\beta}_{w,k'}\cdot\mathbf{x})\text{ for }v\in V_{j,g},\ w\in V_{k,h} \tag{31}$$

with bound

$$(j+k)|a_v||b_w| \prod_{j'=1}^{j} \beta_{v,j'} \prod_{k'=1}^{k} \beta_{w,k'} \text{ for } v \in V_{j,g}, \ w \in V_{k,h} \tag{32}$$

for all terms with $j + k > 0$ and $\tilde{g}(0)\tilde{h}(0)$ for the term with $j = k = 0$.

Distribute the $j + k$ product, and first focus on the $j$ term only. Summing over all the $V_{k,h}$ for all $k$, we get

$$\sum_{k} \sum_{w \in V_{k,h}} j|a_v||b_w| \prod_{j'=1}^{j} \beta_{v,j'} \prod_{k'=1}^{k} \beta_{w,k'} = \tag{33}$$

$$|a_v| \prod_{j'=1}^{j} \beta_{v,j'} \tilde{h}(1).$$

Now summing over the $j$ and $V_{j,g}$ we get $\tilde{g}'(1)\tilde{h}(1)$. If we do the same for the $k$ term, after summing we get $\tilde{g}(1)\tilde{h}'(1)$. These bounds add and we get the desired formula for $\sqrt{M_{gh}}$, which, up to the additional $\tilde{g}(0)\tilde{h}(0)$ term looks is the product rule applied to $\tilde{g}$ and $\tilde{h}$. $\qquad \square$

One immediate application for this corollary is the product of many univariate analytic functions. If we define

$$G(\mathbf{x}) = \prod_{i} g_i(\boldsymbol{\beta}_i \cdot \mathbf{x}) \tag{34}$$

where each of the corresponding $\tilde{g}_i(y)$ have the appropriate convergence properties, then $G$ is efficiently learnable with bound $M_G$ given by

$$\sqrt{M_G} = \frac{d}{dy} \prod_{i} \tilde{g}_i(\beta_i y) \Bigg|_{y=1} + \prod_{i} \tilde{g}_i(0). \tag{35}$$

*Proof of Corollary 2.* Writing out $g(h(\mathbf{x}))$ as a power series in $h(\mathbf{x})$, we have:

$$g(h(\mathbf{x})) = \sum_{k=0}^{\infty} a_k (h(\mathbf{x}))^k. \tag{36}$$

We can bound each term individually, and use the $k$-wise product rule to bound each term of $(h(\mathbf{x}))^k$. Doing this, we have:

$$\sqrt{M_{g \circ h}} = \sum_{k=1}^{\infty} k|a_k|\tilde{h}'(1)\tilde{h}(1)^{k-1} + \sum_{k=0}^{\infty} |a_k|\tilde{h}(0)^k. \tag{37}$$

Factoring out $\tilde{h}'(1)$ from the first term and then evaluating each of the series gets us the desired result. $\qquad \square$

The following corollary considers the case where the function $g(\mathbf{x})$ is low-degree and directly follows from Theorem 8.

**Fact 1.** *The following facts about the tilde function will be useful in our analysis—*

1. *Given a multivariate analytic function $g(\mathbf{x})$ of degree $p$ for $\mathbf{x}$ in the $d$-dimensional unit ball, there is a function $\tilde{g}(y)$ as defined in Theorem 8 such that $g(\mathbf{x})$ is learnable to error $\epsilon$ with $O(p\tilde{g}(1)/\epsilon^2)$ samples.*

2. *The tilde of a sum of two functions is at most the sum of the tilde of each of the functions, i.e. if $f = g + h$ then $\tilde{f}(y) \le \tilde{g}(y) + \tilde{h}(y)$ for $y \ge 0$.*

3. *The tilde of a product of two functions is at most the product of the tilde of each of the functions, i.e. if $f = g \cdot h$ then $\tilde{f}(y) \le \tilde{g}(y)\tilde{h}(y)$ for $y \ge 0$.*

4. If $g(\mathbf{x}) = f(\alpha\mathbf{x})$, then $\tilde{g}(y) \le \tilde{f}(\alpha y)$ for $y \ge 0$.

5. If $g(\mathbf{x}) = f(\mathbf{x} + \mathbf{c})$ for some $\|\mathbf{c}\| \le 1$, then $\tilde{g}(y) \le \tilde{f}(y+1)$ for $y \ge 0$. By combining this with the previous fact, if $g(\mathbf{x}) = f(\alpha(\mathbf{x} - \mathbf{c}))$ for some $\|\mathbf{c}\| \le 1$, then $\tilde{g}(1) \le \tilde{f}(2\alpha)$.

To verify the last part, note that in the definition of $\tilde{g}$ we replace $\langle \boldsymbol{\beta}, \mathbf{x} \rangle$ with $y$. Therefore, we will have an additional $\langle \boldsymbol{\beta}, \mathbf{c} \rangle$ term when we compute the tilde function for $g(\mathbf{x}) = f(\mathbf{x} + \mathbf{c})$. As $\|\mathbf{c}\| \le 1$, the additional term is at most 1.

The following lemma shows how we can approximate the indicator $\mathbf{1}(x > \alpha)$ with a low-degree polynomial if $x$ is at least $\gamma/2$ far away from $\alpha$. We will use this primitive several times to construct low-degree analytic approximations of indicator functions. The result is based on the following simple fact.

**Fact 2.** *If the Taylor series of $g(\mathbf{x})$ is exponentially decreasing, then we can truncate it at degree $O(\log(1/\epsilon))$ to get $\epsilon$ error. We will use this fact to construct low-degree approximations of functions.*

**Lemma 5.** *Given a scalar $x$, let the function*

$$\Phi(x, \gamma, \epsilon, \alpha) = (1/2)\left(1 + erf\left((x-\alpha)c\sqrt{\log(1/\epsilon)}/\gamma\right)\right)$$

*for some constant c. Let $\Phi'(x, \gamma, \epsilon, \alpha)$ be the function $\Phi(x, \gamma, \epsilon, \alpha)$ with its Taylor series truncated at degree $O(\log(1/\epsilon)/\gamma)$. Then for $|\alpha| < 1$,*

$$\Phi'(x, \gamma, \epsilon, \alpha) = \begin{cases} \epsilon & x \le \alpha - \gamma/2, \\ 1 - \epsilon & x \ge \alpha + \gamma/2. \end{cases}$$

*Also, $M_{\Phi'}$ is at most $e^{O((\log(1/\epsilon)/\gamma^2))}$.*

*Proof.* Note that $\Phi(x, \gamma, \epsilon, \alpha)$ is the cumulative distribution function (cdf) of a normal distribution with mean $\alpha$ and standard deviation $O(\gamma/\sqrt{\log(1/\epsilon)})$. Note that at most $\epsilon/100$ of the probability mass of a Gaussian distribution lies more than $O(\sqrt{\log(1/\epsilon)})$ standard deviations away from the mean. Therefore,

$$\Phi(x, \gamma, \epsilon, \alpha) = \begin{cases} \epsilon/100 & x \le \alpha - \gamma/2, \\ 1 - \epsilon/100 & x \ge \alpha + \gamma/2. \end{cases}$$

Note that

$$\text{erf}(x) = \frac{2}{\sqrt{\pi}} \int_0^x e^{-t^2} dt$$

$$= \frac{2}{\sqrt{\pi}} \left(\sum_{i=0}^{\infty} \frac{(-1)^i x^{2i+1}}{i!(2i+1)}\right).$$

Therefore, the coefficients in the Taylor series expansion of $\text{erf}((x-\alpha)c\sqrt{\log(1/\epsilon)}/\gamma))$ in terms of $(x - \alpha)$ are smaller than $\epsilon$ for $i > O(\log(1/\epsilon)/\gamma^2)$ and are geometrically decreasing henceforth. Therefore, we can truncate the Taylor series at degree $O(\log(1/\epsilon)/\gamma^2)$ and still have an $O(\epsilon)$ approximation. Note that for $f(x) = \text{erf}(x)$,

$$\tilde{f}(y) \le \frac{2}{\sqrt{\pi}} \int_0^y e^{t^2} dt \le \frac{2}{\sqrt{\pi}} y e^{y^2} \le e^{O(y^2)}.$$

After shifting by $\alpha$ and scaling by $O(\sqrt{\log(1/\epsilon)}/\gamma)$, we get $\tilde{\Phi}'(y) = e^{O((y+\alpha)^2 \log(1/\epsilon)/\gamma^2)}$. For $x = 1$, this is at most $e^{O(\log(1/\epsilon)/\gamma^2)}$. Hence the result now follows by Fact 1.

$\square$

### A.3 LEARNABILITY OF CLUSTER BASED DECISION NODE

In the informal version of the result for learning cluster based decisions we assumed that the task-codes $\mathbf{c}$ are prefixed to the input datapoints, which we refer to as $\mathbf{x}_{\text{inp}}$. For the formal version of the theorem, we use a small variation. The task code and the input $\mathbf{c}, \mathbf{x}_{\text{inp}}$ gets mapped to $\mathbf{x} = \mathbf{c} + \mathbf{x}_{\text{inp}} \cdot (r/3)$ for

some constant $r < 1/6$. Since $\mathbf{x}_{\text{inp}}$ resides on the unit sphere, $\mathbf{x}$ will be distance at most $(r/3)$ from the center it gets mapped to. Note that the overall function $f$ can be written as follows,

$$f(\mathbf{x}) = \sum_{j=1}^{k} \mathbf{1}\left(\|\mathbf{x} - \mathbf{c}_j\|^2 \leq (r/2)^2\right) f_j\left((\mathbf{x} - \mathbf{c}_j)/(r/3)\right)$$

where $f_j$ is the function corresponding to the center $\mathbf{c}_j$. The main idea will be to show that the indicator function can be expressed as an analytic function.

**Theorem 9.** *(formal version of Theorem 2) Assume that $d \geq 10 \log k$ (otherwise we can pad by extra coordinates to increase the dimensionality). Then we can find $k$ centers in the unit ball which are at least $r$ apart, for some constant $r$. Let*

$$f(\mathbf{x}) = \sum_{j=1}^{k} \mathbf{1}\left(\|\mathbf{x} - \mathbf{c}_j\|^2 \leq (r/2)^2\right) f_j\left((\mathbf{x} - \mathbf{c}_j)/(r/3)\right)$$

*where $f_j$ is the function corresponding to the center $\mathbf{c}_j$. Then if each $f_j$ is a degree $p$ polynomial, $M_f$ of the function $f$ is $p \cdot poly(k/\epsilon) \sum \tilde{f}_j(6/r) \leq p \cdot poly(k/\epsilon)(6/r)^p \sum \tilde{f}_j(1)$.*

*Proof.* Let

$$f_{\text{app}}(\mathbf{x}) = \sum_{j=1}^{k} \Phi'\left(\|\mathbf{x} - \mathbf{c}_j\|^2, (r/2)^2, \epsilon/k, (r/4)^2\right) f_j\left((\mathbf{x} - \mathbf{c}_j)/(r/3)\right)$$

where $\Phi'$ is defined in Lemma 5. Let

$$I_j(\mathbf{x}) = \Phi'(\|\mathbf{x} - \mathbf{c}_j\|^2, (r/2)^2, \epsilon/k, (r/4)^2).$$

The indicator $I_j(\mathbf{x})$ checks if $\|\mathbf{x} - \mathbf{c}_j\|$ is a constant fraction less than $r/2$, or a constant fraction more than $r/2$. Note that if $\mathbf{x}$ is from a different cluster, then $\|\mathbf{x} - \mathbf{c}_j\|$ is at least some constant, and hence $I_j(\mathbf{x})$ is at most $\epsilon/k$. The contribution from $k$ such clusters would be at most $\epsilon$. If $\|\mathbf{x} - \mathbf{c}_j\| < \epsilon/k$, then the indicator is at least $1 - O(\epsilon/k)$. Hence as $f_{\text{app}}$ is an $O(\epsilon)$-approximation to $f$, by Remark 2 it suffices to show learnability of $f_{\text{app}}$.

If $y = \langle \mathbf{x}, \mathbf{c}_j \rangle$ and assuming $\mathbf{x}$ and the centers $\mathbf{c}_j$ are all on unit sphere,

$$\tilde{I}_j(y) = \tilde{\Phi}'(2 + 2y, r/3, \epsilon/k, r/3) \leq e^{O(\log(k/\epsilon))} = \text{poly}(k/\epsilon).$$

By Fact 1,

$$\tilde{f}(y) \leq \text{poly}(k/\epsilon) \sum_j \tilde{f}_j(6/r).$$

As $f_j$ are at most degree $p$,

$$\tilde{f}(y) \leq \text{poly}(k/\epsilon) \sum_j \tilde{f}_j(6/r) \leq p \cdot \text{poly}(k/\epsilon)(6/r)^p \sum_j \tilde{f}_j(1).$$

$\square$

**Corollary 4.** *The previous theorem implies that we can also learn $f$ where $f$ is a lookup table with $M_f = poly(k/\epsilon)$, as long as the keys $c_i$ are well separated. Note that as long as the keys $c_i$ are distinct (for example, names) we can hash them to random vectors on a sphere so that they are all well-separated.*

Note that the indicator function for the informal version of Theorem 9 stated in the main body is the same as that for the lookup table in Corollary 4. Therefore, the informal version of Theorem 9 follows as a Corollary of Theorem 9.

A.4  LEARNABILITY OF FUNCTIONS DEFINED ON LEAVES OF A DECISION TREE

We consider decision trees on inputs drawn from $\{-1, 1\}^d$. We show that such a decision tree $g$ can be learnt with $M_g \leq O(d^h)$. From this section onwards, we view the combined input $\mathbf{c}, \mathbf{x}$ as $\mathbf{x}$.

The decision tree $g$ can be written as follows,

$$g(\mathbf{x}) = \sum_j I_j(\mathbf{x})v_j,$$

where the summation runs over all the leaves, $I_j(\mathbf{x})$ is the indicator function for leaf $j$, and $v_j \in [-1, 1]$ is the constant value on the leaf $j$. We scale the inputs by $\sqrt{d}$ to make them lie on the unit sphere, and hence each coordinate of $\mathbf{x}$ is either $\pm 1/\sqrt{d}$.

Let the total number of leaves in the decision tree be $B$. The decision tree indicator function of the $j$-th leaf can be written as the product over the path of all internal decision nodes. Let $j_l$ be variable at the $l$-th decision node on the path used by the $j$-th leaf. We can write,

$$I_j(\mathbf{x}) = \prod_l \left( a_{j_l} x_{j_l} + b_{j_l} \right),$$

where each $x_{j_l} \in \{-1/\sqrt{d}, 1/\sqrt{d}\}$ and $a_{j_l} \in \{-\sqrt{d}/2, \sqrt{d}/2\}$ and $b_{j_l} \in \{-1/2, 1/2\}$. Note that the values of $a_{j_l}$ and $b_{j,l}$ are chosen depending on whether the path for the $j$-th leaf choses the left child or the right child at the $l$-th decision variable. For ease of exposition, the following theorem is stated for the case where the leaf functions are constant functions, and the case where there are some analytic functions at the leaves also follows in the same way.

**Theorem 10.** *If a function is given by $g(\mathbf{x}) = \sum_{j=1}^{B} I_j(\mathbf{x})v_j$, where $I_j(\mathbf{x})$ is a leaf indicator function in the above form, with tree depth $h$, then $M_g$ is at most $O(d^h)$.*

*Proof.* Note that

$$\tilde{g}(y) \leq \sum \tilde{I}_j(y)|v_j|$$
$$\leq \sum \prod_l \left( \sqrt{d}y/2 + 1/2 \right)$$
$$\implies \tilde{g}(1) \leq 2^h(\sqrt{d}/2 + 1/2)^h \leq d^h.$$

As the degree of $g$ is at most $h$, therefore $M_g \leq h\tilde{g}(1) \leq hd^h$. $\qquad \square$

**Remark 3.** *Note that by Theorem 10 we need $O\left((\log k)^{\log k}\epsilon^{-2}\right)$ samples to learn a lookup table based on a decision tree. On the other hand, by Corollary 4 we need $\text{poly}(k/\epsilon)$ samples to learn a lookup table using cluster based decision nodes. This shows that using a hash function to obtain a random $O(\log k)$ bit encoding of the indexes for the $k$ lookups is more efficient than using a fixed $\log k$ length encoding for the $k$ lookups.*

We also prove a corresponding lower bound in Theorem 14 which shows that $d^{\Omega(h)}$ samples are necessary to learn decision trees of depth $h$.

We will now consider decision trees where the branching is based on the inner product of $\mathbf{x}$ with some direction $\boldsymbol{\beta}_{j,l}$. Assume that there is a constant gap for each decision split, then the decision tree indicator function can be written as,

$$I_j(\mathbf{x}) = \prod_l \mathbf{1}(\langle \mathbf{x}, \boldsymbol{\beta}_{j,l} \rangle > \alpha_{j,l}).$$

**Theorem 11.** *(formal version of Theorem 4) A decision tree of depth $h$ where every node partitions in a certain direction with margin $\gamma$ can be written as $g(\mathbf{x}) = \sum_{j=1}^{B} I_j(\mathbf{x})f_j(\mathbf{x})$, then the final*

$$M_g = e^{O(h \log(1/\epsilon)/\gamma^2)}(p + h \log 1/\epsilon) \sum \tilde{f}_j(1),$$

*where $p$ is the maximum degree of $f_j$.*

*Proof.* Define $g_{\mathrm{app}}$,

$$g_{\mathrm{app}}(\mathbf{x}) = \sum_{j=1}^{B} \Pi_l \Phi'(\langle \mathbf{x}, \boldsymbol{\beta}_{j,l} \rangle, \gamma, \epsilon/h, \alpha_{j,l}) f_j(\mathbf{x})$$

where $\Phi'$ is as defined in Lemma 5. Note that for all $y = 1$,

$$\tilde{\Phi}'(1, \gamma, \epsilon/h, \alpha_{j,l}) \le e^{O(\log(1/\epsilon)/\gamma^2)}.$$

Therefore,

$$\tilde{g}_{\mathrm{app}}(1) \le \sum_{j=1}^{B} \Pi_l \tilde{\Phi}'(1, \gamma, \epsilon/h, \alpha_{j,l}) \tilde{f}_j(1),$$

$$\le e^{O(\log(1/\epsilon)/\gamma^2)} \sum \tilde{f}_j(1).$$

Note that the degree of $g_{\mathrm{app}}$ is at most $O(p + h \log(1/\epsilon)/\gamma^2)$. Therefore,

$$M_{g_{\mathrm{app}}} \le e^{O(h \log(1/\epsilon)/\gamma^2)} (p + h \log(1/\epsilon)/\gamma^2) \sum \tilde{f}_j(1).$$

By Remark 2, learnability of $g$ follows from the learnability of its analytic approximation $g_{\mathrm{app}}$. $\square$

## A.5 GENERALIZED DECISION PROGRAM

In this section, instead a decision tree, we will consider a circuit with fan-out 1, where each gate (node) evaluates some function of the values returned by its children and the input $\mathbf{x}$. A decision tree is a special case of such circuits in which the gates are all switches.

So far, the function outputs were univariate but we will now generalize and allow multivariate (vector) outputs as well. Hence the functions can now evaluate and return data structures, represented by vectors. We assume that each output is at most $d$ dimensional and lies in the unit ball.

**Definition 4.** *For a multivariate output function $f$, we define $\tilde{f}(y)$ as the sum of $\tilde{f}_i(y)$ for each of the output coordinates $f_i$.*

**Remark 4.** *Theorem 9 , 10 and 11 extend to the multivariate output case. Note that if each of the individual functions has degree at most $p$, then the sample complexity for learning the multivariate output $f$ is at most $O(p\tilde{f}(1)/\epsilon^2))$ (where the multivariate tilde function is defined in Definition 4).*

We now define a generalized decision program and the class of functions that we support.

**Definition 5.** *We define a generalized decision program to be a circuit with fan-out 1 (i.e., a tree topology) where each gate evaluates a function of the values returned by its children and the input $\mathbf{x}$, and the root node evaluates the final output. All gates, including those at the leaves, have access to the input $\mathbf{x}$. We support the following gate operations. Let $h$ be the output of a gate, let each gate have at most $k$ children, and let $\{f_1, \ldots, f_k\}$ be the outputs of its children.*

1. *Any analytic function of the child gates of degree at most $p$, including sum $h = \sum_{i=1}^{k} f_i$ and product of $p$ terms $h = \Pi_{i=1}^{p} f_i$.*

2. *Margin based switch (decision) gate with children $\{f_1, f_2\}$, some constant margin $\gamma$, vector $\boldsymbol{\beta}$ and constant $\alpha$,*

$$h = \begin{cases} f_1 & \text{if } \langle \boldsymbol{\beta}, \mathbf{x} \rangle - \alpha \le -\gamma/2, \\ f_2 & \text{if } \langle \boldsymbol{\beta}, \mathbf{x} \rangle - \alpha \ge \gamma/2. \end{cases}$$

3. *Cluster based switch gate with $k$ centers $\{\mathbf{c}^{(1)}, \ldots, \mathbf{c}^{(k)}\}$, with separation $r$ for some constant $r$, and the output is $f_i$ if $\|\mathbf{x} - \mathbf{c}^{(i)}\| \le r/3$. A special case of this is a look-up table which returns value $v_i$ if $\mathbf{x} = \mathbf{c}^{(i)}$, and 0 if $\mathbf{x}$ does not match any of the centers.*

4. *Create a data structure out of separate fields by concatenation such as constructing a tuple $[f_1, \ldots, f_k]$ which creates a single data structure out of its children, or extract a field out of a data structure.*

5. *Given a table $T$ with $k$ entries $\{r_1, \ldots, r_k\}$, a Boolean-valued function $p$ and an analytic function $f$, SQL queries of the form* `SELECT SUM f(r_i), WHERE p(r_i, x)`. *Here, we assume that $f$ has bounded value and $p$ can be approximated by an analytic function of degree at most $p$.*

6. *Compositions of functions, $h(\mathbf{x}) = f(g(\mathbf{x}))$.*

First, we note that all of the above operators can be approximated by low-degree polynomials.

**Claim 1.** *If $p \leq O(\log(k/\epsilon))$, each of the above operators in the generalized decision program can be expressed as a polynomial of degree at most $O(\log(k/\epsilon))$, where $k$ is maximum out-degree of any of the nodes.*

**Remark 5.** *Note that for the SQL query, we can also approximate other aggregation operators apart from SUM, such as MAX or MIN. For example, to approximate MAX of $x_1, \ldots, x_k$ up to $\epsilon$ where the input lies between $[0, 1]$ we can first write it as*

$$MAX(x_1, \ldots, x_k) = \epsilon \sum_j \mathbf{1} \left( \sum_i (\mathbf{1}(x_i > \epsilon j) > 1/2) \right),$$

*and then approximate the indicators by analytic functions.*

Lemma 6 shows how we can compute the tilde function of the generalized decision program.

**Lemma 6.** *The tilde function for a generalized decision program can be computed recursively with the following steps:*

1. *For a sum gate $h = f + g$, $\tilde{h}(y) = \tilde{f}(y) + \tilde{g}(y)$.*

2. *For a product gate, $h = f.g$, $\tilde{h}(y) = \tilde{f}(y) \cdot \tilde{g}(y)$.*

3. *For a margin based decision gate (switch) with children $f$ and $g$, $h = I_{left} f + (1 - I_{left}) g$ and $\tilde{h}(y) = \tilde{I}_{left}(\tilde{f}(y) + \tilde{g}(y)) + \tilde{g}(y)$. Here $I_{left}$ is the indicator for the case where the left child is chosen.*

4. *For cluster based decision gate (switch) with children $\{f_1, ..., f_k\}$, $\tilde{h}(y) \leq \sum_i \tilde{I}_i \tilde{f}_i(6y/r)$. Here $I_i$ is the indicator for the cluster corresponding to the $i$-th child.*

5. *For a look-up table with $k$ key-values, $\tilde{h}(y) \leq k\tilde{I}(y)$ as long as the $\ell_1$ norm of each key-value is at most 1.*

6. *Creating a data structure out of separate fields can be done by concatenation, and $\tilde{h}$ for the result is at most sum of the original tilde functions. Extracting a field out of a data structure can also be done in the same way.*

7. *Given an analytic function $f$ and a Boolean function $p$, for a SQL operator $h$ over a table $T$ with $k$ entries $\{r_1, \ldots, r_k\}$ representing* `SELECT SUM f(r_i), WHERE p(r_i, x)`, *or in other words $h = \sum_i f(r_i) p(r_i, x)$, $\tilde{h}(y) \leq \sum_i \tilde{I}_{p, r_i}(y)$, where $I_{p, r_i}$ is the indicator for $p(r_i, x)$. For example, $x$ here can denote some threshold value to be applied to a column of the table, or selecting some subset of entries (in Fig. 1, $x$ is the zip-code).*

8. *For $h(\mathbf{x}) = f(g(\mathbf{x}))$, $\tilde{h}(y) \leq \tilde{f}(\tilde{g}(y))$.*

All except for the last part of the above Lemma directly follow from the results in the previous sub-section. Below, we prove the result for the last part regarding function compositions.

**Lemma 7.** *Assume that all functions have input and output dimension at most $d$. If $f$ and $g$ are two functions with degree at most $p_1$ and $p_2$, then $h(\mathbf{x}) = f(g(\mathbf{x}))$ has degree at most $p_1 p_2$ and $\tilde{h}(y) \leq \tilde{f}(\tilde{g}(y))$.*

*Proof.* Note that this follows if $f$ and $g$ are both scalar outputs and inputs. Let $g(\mathbf{x}) = (g_1(\mathbf{x}), ..., g_d(\mathbf{x}))$. Let us begin with the case where $f = \langle \boldsymbol{\beta}, \mathbf{x} \rangle$, where $\|\boldsymbol{\beta}\| = 1$. Then

$\tilde{h}(y) = \sum_i |\beta_i| \tilde{g}_i(y) \leq \sum_i \tilde{g}_i(y) \leq \tilde{g}(y)$. When $f = \Pi_{i=1}^{p_1} \langle \boldsymbol{\beta}_i, \mathbf{x} \rangle$, $\tilde{h}(y) \leq \tilde{g}(y)^{p_1} \leq \tilde{f}(\tilde{g}(y))$. The same argument works when we take a linear combination, and also for a multivariate function $f$ (as $\tilde{f}$ for a multivariate $f$ is the summation of individual $\tilde{f}_i$, by definition). □

We now present our result for learning generalized decision programs.

**Theorem 12.** *Let the in-degree of any gate be at most $k$. The sample complexity for learning the following classes of generalized decision programs is as follows:*

1. *If every gate is either a decision node with margin $\gamma$, a sum gate, or a lookup of size at most $k$, then $M_g \leq e^{O(h \log(1/\epsilon)/\gamma^2)} k^{O(h)}$.*

2. *For some constant $C$, if there are at most $C$ product gates with degree at most $C$, and every other gate is a decision gate with margin $\gamma$ or a sum gate with constant functions at the leaves, then $M_g \leq e^{O(h \log(1/\epsilon)/\gamma^2)}$.*

3. *Given a function $f$ and a Boolean function $p$ which can be approximated by a polynomial of degree at most $O(\log(k/\epsilon))$, for a SQL operator $g$ over a table $T$ with $k$ entries $\{r_1, \ldots, r_k\}$ representing* `SELECT SUM f(r_i), WHERE p(r_i, x)`*, $M_g \leq \sum_i \tilde{I}_{p,r_i}(1)$.*

4. *Let the function at every gate be an analytic function $f$ of degree at most $p$ and the sum of the coefficients of $f$ is upper bounded by $c^p$ for some constant $c$. Then note that $\tilde{f}(y) \leq (cy)^p$ for $y \geq 1$. Therefore, the final function $\tilde{g}(y) \leq (cky)^{p^h}$ and hence $M_g \leq (ck)^{p^h}$.*

*Proof.* The first three claims can be obtained using Lemma 6.

For the final claim, consider the final polynomial obtained by expanding the function at each gate in a bottom-up way. We will upper bound $\tilde{g}(y)$ for the overall function $g$ corresponding to the generalized decision program. $\tilde{g}(y)$ can be upper bounded by starting with $\tilde{f}(y)$ for the leaf nodes $f$. For any internal gate $i$, let $g_i(x) = f_i(f_{j_1}(x), \ldots, f_{j_p}(x))$ where $f_{j_t}$ are the outputs of the children of the gate $i$. We recursively compute $\tilde{g}_i(y) = \tilde{f}_i(\sum_l \tilde{f}_{j_l}(y))$. Therefore, for a gate with $k$ children $\tilde{g}_i(y) \leq (c \sum_l \tilde{g}_{j_l}(y))^p$. Therefore, for the root gate $g_0$, $\tilde{g}_0(y) \leq (cky)^{p^h}$. □

**Remark 6.** *Note that the dependence on $h$ is doubly exponential. We show a corresponding lower bound in Theorem 15 that this is necessary.*

Theorem 12 implies that we can learn programs such as the following formal version of Fig. 1 (right)—which involves analytic functions, SQL queries, data structures, and table look-up.

**Example 1.** *Consider the following program:*

```
class Person{
    string name;
    Address address;
    int income;
    public string get_zip_code(){
        return address.zip_code;
    }
    init(input_name, input_address, input_income){
        name = input_name;
        address = input_address;
        income = input_income;
    }
}
class Address{
    int street_number;
    string street_name;
    string city;
    string state;
    string zip_code;
```

```
    public string get_zip_code(){
        return zip_code;
    }
    init(...){
        ... # function to create new object with input values
    }
}
dictionary name_to_address_table;
dictionary zip_code_to_lat_long; #maps zip_code to tuple of (latitute, longitude)

boolean in_same_zip_code(Person A, Person B){
    return A.get_zip_code() == B.get_zip_code();
}

float get_straight_line_distance(Person A, Person B){
    lat_longA =  zip_code_to_lat_long[A.get_zip_code()];
    lat_longB =  zip_code_to_lat_long[B.get_zip_code()];
    return euclidean_distance(lat_longA, lat_longB);
}

float avg_income_zip_code(string zip_code){
    construct SQL table T with income, zip_code from name_to_address_table;
    return output of SQL query "SELECT AVG(INCOME) FROM T WHERE ZIP_CODE=zip_code"
}
```

The following claim follows from Theorem 12.

**Claim 2.** *The above classes and functions can be implemented and learnt using* $(k/\epsilon)^{O(\log(1/\epsilon))}$
*samples, where the tables are of size at most* $k$.

*Proof.* We begin with the `in_same_zip_code()` function. Note that this is a special case of the cluster based functions. As in Corollary 4 all attributes such as zip-code are appropriately hashed such that they are well-separated. We can now test equality by doing an indicator function for a ball around the zip-code of Person A. The indicator function for a ball can be approximated by a low-degree polynomial as in the cluster-based branching results in Theorem 9. As the total number of individuals is at most $k$, therefore by Theorem 9 the sample complexity is at most $\text{poly}(k/\epsilon)$.

For the `avg_income_zip_code()` function, we use the SQL query result in Theorem 12. Note that the indicators are testing equality in the case of our program, and hence as in the previous case we can use the cluster-based branching result in Theorem 9 to approximate these indicators by polynomial functions, to obtain a sample complexity of $\text{poly}(k/\epsilon)$.

Finally, we argue that we can learn the `get_straight_line_distance()` function. Here, we are composing two functions $f$ and $(g_1, g_2)$ where $f$ is the distance function and $(g_1, g_2)$ are the lookups for the latitude and longitude for Person A and B. By Corollary 4, the lookups have $\tilde{g}_i(1) \leq \text{poly}(k/\epsilon)$. By part 6 of Lemma 6, the tilde for the concatenation is the sum of the tilde for the individual functions. For computing the Euclidean distance $\sqrt{\sum(x_i - y_i)^2}$, note that the square root function does not have a Taylor series defined at 0. However, we can use the same analysis as in the proof for learning the $1/x$ function in the gravitational law (see Appendix B.1) to get a polynomial of degree at most $O(\log(1/\epsilon))$, and hence $\tilde{f}(y) \leq (O(y))^{\log(1/\epsilon)}$. Thus using the composition rule in Lemma 6, the sample complexity is $(k/\epsilon)^{O(\log(1/\epsilon))}$.

$\square$

# B  LEARNING DYNAMICAL SYSTEMS

## B.1  GRAVITATIONAL FORCE LAW

We can use the product and chain rules to show that many functions important in scientific applications can be efficiently learnable. This is true even when the function has a singularity. As an example demonstrating both, we prove the following bound on learning Newton's law of gravitation:

**Theorem 13.** *Consider a system of $k$ bodies with positions $\mathbf{x}_i \in \mathbb{R}^3$ and masses $m_i$, interacting via the force:*

$$\mathbf{F}_i = \sum_{j \neq i} \frac{m_i m_j}{r_{ij}^3}(\mathbf{x}_j - \mathbf{x}_i) \tag{38}$$

*where $r_{ij} \equiv ||\mathbf{x}_i - \mathbf{x}_j||$. We assume that $R = r_{max}/r_{min}$, the ratio between the largest and smallest pairwise distance between any two bodies, is constant. Suppose the $m_i$ have been rescaled to be between $0$ and $1$. Then the force law is efficiently learnable in the sense of Definition 3 using the modified ReLU kernel to generalization error less than $\epsilon$ using $k^{O(\ln(k/\epsilon))}$ samples.*

*Proof.* We will prove learning bounds for each component of $F$ separately, showing efficient learning with probability greater than $1 - \delta/3k$. Then, using the union bound, the probability of simultaneously learning all the components efficiently will be $1 - \delta$.

There are two levels of approximation: first, we will construct a function which is within $\epsilon/2$ of the original force law, but more learnable. Secondly, we will prove bounds on learning that function to within error $\epsilon/2$.

We first rescale the vector of collective $\{\mathbf{x}_i\}$ so that their collective length is at most 1. In these new units, this gives us $r_{max}^2 \leq \frac{2}{k}$. The first component of the force on $\mathbf{x}_1$ can be written as:

$$(\mathbf{F}_1)_1 = \sum_{j=2}^{k} \frac{m_1 m_j}{r_{1j}^2} \frac{((\mathbf{x}_j)_1 - (\mathbf{x}_1)_1)}{r_{1j}}. \tag{39}$$

If we find a bound $\sqrt{M_f}$ for an individual contribution $f$ to the force, we can get a bound on the total $\sqrt{M_F} = (k-1)\sqrt{M_f}$. Consider an individual force term in the sum. The force has a singularity at $r_{1j} = 0$. In addition, the function $r_{1j}$ itself is non-analytic due to the branch cut at 0.

We instead will approximate the force law with a finite power series in $r_{1j}^2$, and get bounds on learning said power series. The power series representation of $(1-x)^{-3/2}$ is $\sum_{n=0}^{\infty} \frac{(2n+1)!!}{(2n)!!} x^n$. If we approximate the function with $d$ terms, the error can be bounded using Taylor's theorem. The Lagrange form of the error gives us the bound

$$\left| \frac{1}{(1-x)^{3/2}} - \sum_{n=0}^{d} \frac{(2n+1)!!}{(2n)!!} x^n \right| \leq \frac{\sqrt{\pi d}|x|^{d+1}}{(1-|x|)^{5/2+d}} \tag{40}$$

where we use $\frac{(2n+1)!!}{(2n)!!} \approx \sqrt{\pi n}$ for large $n$. We can use the above expansion by rewriting

$$r_{1j}^{-3} = a^{-3}(1 - (1 - r_{1j}^2/a^2))^{-3/2} \tag{41}$$

for some shift $a$. Approximation with $f_d(r_{1j}^2)$, the first $d$ terms of the power series in $(1 - r_{1j}^2/a^2)$ gives us the error:

$$|f_d(r_{1j}^2) - r_{1j}^{-3}| \leq \frac{\sqrt{\pi d}|1 - r_{1j}^2/a^2|^{d+1}}{a^3(1 - |1 - r_{1j}^2/a^2|)^{5/2+d}} \tag{42}$$

which we want to be small over the range $r_{min} \leq r_{1j} \leq r_{max}$.

The bound is optimized when it takes the same value at $r_{min}$ and $r_{max}$, so we set $a^2 = (r_{min}^2 + r_{max}^2)/2$. In the limit that $r_{max} \gg r_{min}$, where learning is most difficult, the bound becomes

$$|f_d(r_{1j}^2) - r_{1j}^{-3}| \leq \frac{\sqrt{8\pi d}}{r_{max}^3} \left(R^2/2\right)^{5/2+d} e^{-2(d+1)/R^2} \tag{43}$$

where $R = r_{max}/r_{min}$, which is constant by assumption.

In order to estimate an individual contribution to the force force to error $\epsilon/2k$ (so the total error is $\epsilon/2$), we must have:

$$m_1 m_j r_{max} |f_d(r_{1j}) - r_{1j}^{-3}| \leq \frac{\epsilon}{2k} \tag{44}$$

This allows us to choose the smallest $d$ which gives us this error. Taking the logarithm of both sides, we have:

$$\frac{1}{2} \ln(d) - (5/2 + d) \ln\left(2/R^2\right) - 2(d+1)/R^2 \leq \ln(\epsilon/k^2). \tag{45}$$

where we use that $r_{max}^2 \leq 2/k$ after rescaling. The choice $d \geq R^2 \ln(k^2/\epsilon)$ ensures error less than $\epsilon/2k$ per term.

Using this approximation, we can use the product and chain rules to get learning bounds on the force law. We can write the approximation

$$F_\epsilon(\mathbf{x}) = \sum_{j \neq 1} m_1 m_j f_d(h_j(\mathbf{x})) k_j(\mathbf{x}) \tag{46}$$

where $h_j(\mathbf{x}) = ||\mathbf{x}_1 - \mathbf{x}_j||$ and $k_j(\mathbf{x}) = (\mathbf{x}_1)_1 - (\mathbf{x}_j)_j$ The number of samples needed for efficient learning is bounded by $\sqrt{M_{F_\epsilon}} = \frac{\sqrt{8k}}{r_{max}^3} A_{F_\epsilon}$, for

$$A_{F_\epsilon} = \tilde{f}_d'(\tilde{h}(1))\tilde{h}'(1)\tilde{k}(1) + \tilde{f}_d(\tilde{h}(1))\tilde{k}'(1) \tag{47}$$

with

$$\tilde{k}(y) = \sqrt{2}y, \ \tilde{h}(y) = 6y^2, \ \tilde{f}_d(y) = \sqrt{\pi d}(1 + y/a^2)^d. \tag{48}$$

Evaluating, we have

$$A_{F_\epsilon} = \sqrt{2\pi d}\left(1 + \frac{12}{r_{max}^2}\right)^d + \sqrt{\pi d^3}\left(1 + \frac{12}{r_{max}^2}\right)^{d-1} \tag{49}$$

which, after using $r_{max}^2 \leq 2/k$ and $d = R^2 \ln(k^2/\epsilon)$ gives us the bound

$$\sqrt{M_{F_\epsilon}} \leq k^{-1/2} \left(R^2 \ln(k^2/\epsilon)\right)^{3/2} (24k)^{R^2 \ln(k^2/\epsilon)}. \tag{50}$$

The asymptotic behavior is

$$\sqrt{M_{F_\epsilon}} = k^{O(\ln(k/\epsilon))} \tag{51}$$

since $R$ is bounded.

We can therefore learn an $\epsilon/2$-approximation of one component of $\mathbf{F}_1$, with probability at least $1 - \delta/3k$ and error $\epsilon/2$ with $O(4(M_{F_\epsilon} + \log(3k/\delta))/\epsilon^2)$ samples. Therefore, we can learn $\mathbf{F}_1$ to error $\epsilon$ with the same number of samples. Using a union bound, with probability at least $1 - \delta$ we can simultaneously learn all components of all $\{\mathbf{F}_i\}$ with that number of samples. $\square$

We note that since the cutoff of the power series at $d(\epsilon) = O(R^2 \ln(k^2/\epsilon))$ dominates the bound, we can easily compute learning bounds for other power-series kernels as well. If the $d$th power series coefficient of the kernel is $b_d$, then the bound on $\sqrt{M_{F_\epsilon}}$ is increased by $(d(\epsilon)^2 b_{d(\epsilon)})^{-1/2}$. For example, for the Gaussian kernel, since $b_d^{-1/2} = \sqrt{d!}$, the bound becomes

$$\sqrt{M_{F_\epsilon}} = (R^2 \ln(k^2/\epsilon)k)^{O(\ln(k/\epsilon))} \tag{52}$$

which increases the exponent of $k$ by a factor of $\ln(R^2 \ln(k^2/\epsilon))$.

## B.2 EMPIRICAL CONFIRMATION OF LEARNING BOUNDS

We empirically validated our analytical learning bounds by training models to learn the gravitational force function for $k$ bodies (with $k$ ranging from 5 to 400) in a $3-$dimensional space. We created synthetic datasets by randomly drawing $k$ points from $[0, 1]^3$ corresponding to the location of $k$ bodies, and compute the gravitational force (according to Figure 1) on a target body also drawn randomly from $[0, 1]^3$. To avoid singularities, we ensured a minimum distance of $0.1$ between the target body and the other bodies (corresponding to the choice $R = 10$). As predicted by the theory,

none of the models learn well if $R$ is not fixed. We randomly drew the masses corresponding to the $k + 1$ bodies from $[0, 10]$. We generated 5 million such examples - each example with $4(k + 1)$ features corresponding to the location and mass of each of the bodies, and a single label corresponding to the gravitational force $F$ on the target body along the $x$-axis. We held out $10\%$ of the dataset as test data to compute the root mean square error (RMSE) in prediction. We trained three different neural networks on this data, corresponding to various kernels we analyzed in the previous section:

1. A wide one hidden-layer ReLU network (corresponding to the ReLU NTK kernel).
2. A wide one hidden-layer ReLU network with a constant bias feature added to the input (corresponding to the NTK kernel).
3. A wide one hidden-layer network with exponential activation function, where only the top layer of the network is trained (corresponding to the Gaussian kernel).

We used a hidden layer of width 1000 for all the networks, as we observed that increasing the network width further did not improve results significantly. All the hidden layer weights were initialized randomly.

In Figure 5 we show the normalized RMSE (RMSE/$[F_{max} - F_{min}]$) for each of the neural networks for different values of the number of bodies $k$.

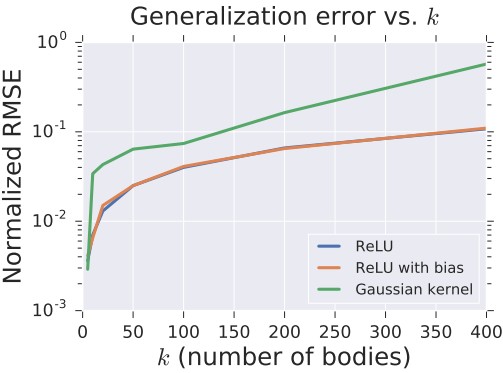

Figure 5: RMSE vs number of bodies $k$ for learning gravitational force law for different kernels. Normalized by the range $F_{max} - F_{min}$ of the forces. Gaussian kernels learn worse than ReLU at large $k$.

All three networks are able to learn the gravitational force equation with small normalized RMSE for hundreds of bodies. Both the ReLU network and ReLU with bias outperform the network corresponding to the Gaussian kernel (in terms of RMSE) as $k$ increases. In particular, the Gaussian kernel learning seems to quickly degrade at around 400 bodies, with a normalized RMSE exceeding $50\%$. This is consistent with the learning bounds for these kernels in Section A.2, and suggests that those bounds may in fact be useful to compare the performances of different networks in practice.

We did not, however, observe much difference in the performance of the ReLU network when adding a bias to the input, which suggests that the inability to get an analytical bound due to only even powers in the ReLU NTK kernel might be a shortcoming of the proof technique, rather than a property which fundamentally limits the model.

## C LOWER BOUNDS

First, we show an exponential dependence on the depth $h$ is necessary for learning decision trees. The result depends on the hardness of solving parity with noise.

**Conjecture 1.** *(hardness of parity with noise) Let $\mathbf{a}, \mathbf{x} \in \{0, 1\}^d$ be d-dimensional Boolean vectors. In the parity with noise problem, we are given noisy inner products modulo 2 of the unknown vector $\mathbf{x}$ with the examples $\mathbf{a}_i$, i.e. $b_i = \langle \mathbf{a}_i, \mathbf{x} \rangle + \eta_i \mod 2$ where $\eta_i$ is a Binomial random variable which is 1 with probability 0.1. Then any algorithm for finding $\mathbf{x}$ needs at least $2^{\tilde{\Omega}(d)}$ time or examples*

*(where $\tilde{\Omega}$ hides poly-logarithmic factors in d). Similarly, if $\mathbf{x}$ is given to be s-sparse for $s \ll d$, then any algorithm for finding $\mathbf{x}$ needs at least $d^{\Omega(s)}$ time or examples.*

Note that the hardness of learning parity with noise is a standard assumption in computational learning theory and forms the basis of many cryptographic protocols (Regev, 2009). The best known algorithm for solving parity needs $2^{O(d/\log d)}$ time and examples (Blum et al., 2003). Learning parities is also known to provably require $2^{\Omega(d)}$ samples for the class of algorithm known as *statistical query algorithms*—these are algorithms are only allowed to obtain estimates of statistical properties of the examples but cannot see the examples themselves (Kearns, 1998). Note that the usual stochastic algorithms for training neural networks such as SGD can be implemented in the statistical query model (Song et al., 2017). Similar hardness result are conjectured for the problem of learning sparse parity with noise, and the best known algorithm runs in time $d^{\Omega(s)}$ (Valiant, 2015).

Based on the hardness of parity with noise, we show that exponential dependence on the depth for learning decision trees is necessary.

**Theorem 14.** *Conditioned on the hardness of the sparse parity with noise problem, any algorithm for learning decision trees of depth $h$ needs at least $d^{\Omega(h)}$ time or examples.*

*Proof.* Note that we can represent a parity with noise problem where the answer is $h$-sparse by a decision tree of depth $h$ where the leaves represent the solutions to the parity problem. The result then follows by the hardness of the sparse parity with noise problem. $\square$

We also show that the doubly exponential dependence on the depth for learning generalized decision programs is necessary.

**Theorem 15.** *Learning a generalized decision program which is a binary tree of depth $h$ using stochastic gradient descent requires at least $2^{2^{\Omega(h)}}$ examples. Conditioned on the hardness of learning noisy parities, any algorithm for learning a generalized program of depth $h$ needs at least $2^{2^{\tilde{\Omega}(h)}}$ time or examples (where $\tilde{\Omega}$ hides poly-logarithmic factors in $h$).*

*Proof.* Note that a generalized decision program of depth $h$ can encode a parity function over $D = 2^h$ bits. Any statistical query algorithm to learn a parity over $D$ bits needs at least $2^{\Omega(D)}$ samples. As stochastic gradient descent can be implemented in the statistical query model, hence the bound for stochastic gradient descent follows.

To prove the general lower bound, note that a generalized decision program of depth $h$ can also encode a *noisy* parity function over $D = 2^h$ bits. Conditioned on the hardness of parity with noise, any algorithm for learning noisy parities needs at least $2^{\tilde{\Omega}(D)}$ samples. Hence the bound for general algorithms also follows. $\square$

In our framework, we assume that all the underlying functions that we learn are analytic, or have an analytic approximation. It is natural to ask if such an assumption is necessary. Next, we show that learning even simple compositions of functions such as their sum is not possible without some assumptions on the individual functions.

**Lemma 8.** *There exists function classes $F_1$ and $F_2$ which can be learnt efficiently but for every $f_1 \in F_1$ there exists $f_2 \in F_2$ such that $f_1 + f_2$ is hard to learn (conditioned on the hardness of learning parity with noise)*

*Proof.* Both $f_1$ and $f_2$ are modifications of the parity with noise problem. The input in both cases is $\mathbf{x} \in \{0,1\}^d$. Let $\boldsymbol{\beta}$ be the solution to the noisy parity problem. The output for the function class $F_1$ is $[\boldsymbol{\beta}, y]$, where $y$ is the value of the noisy parity for the input. The output for the function class $F_2$ is $[-\boldsymbol{\beta}, y]$, where $y$ is again the value of the noisy parity for the input. Note that $F_1$ and $F_2$ are trivial to learn, as the solution $\boldsymbol{\beta}$ to the noisy parity problem is already a part of the output. For any $f_1 \in F_1$, choose $f_2 \in F_2$ to be the function with the same vector $\boldsymbol{\beta}$. Note that conditioned on the hardness of learning parity with noise, $f_1 + f_2$ is hard to learn. $\square$

### C.1 Lower bounds for learning any analytic function

In this section, we show that there is a lower bound on the Rademacher complexity $\bar{\mathbf{y}}^T \bar{H}^{-1} \mathbf{y}$ based on the coefficients in the polynomial expansion of the $\tilde{g}$ function. Hence the $\tilde{g}$ function characterizes the complexity of learning $g$.

For any $J = (J_1, \ldots, J_n) \in \mathbb{N}^n$, write a monomial $X_J = x_1^{J_1} \ldots x_n^{J_n}$. Define $|J| = \sum_k J_k$. For a polynomial $p(x) = \sum_J a_J x_J$, where $a_J \in \mathbb{C}$, its degree $\deg(p) = \max_{a_J \neq 0} |J|$. The following fact shows that monomials form an orthogonal basis over the unit circle in the complex plane.

**Fact 3.** $\langle X_J, X_{J'} \rangle_{\mathbb{C}^n} = 1$ *if $J = J'$ and 0 otherwise (here, $\langle \cdot, \cdot \rangle_{\mathbb{C}^n}$ denotes the inner product over the unit circle in the complex plane).*

Note that according to Theorem 7 the sample complexity for learning $g(x)$ depends on $\tilde{g}'(1) = \sum_j j|a_j|$, and hence is the $\ell_1$ norm of the derivative. The following Lemma shows that this is tight in the sense that $\Omega(\sum_j j a_j^2)$ samples or the $\ell_2$ norm of the derivative are necessary for learning $g(x)$.

For any variable $x$ let $\bar{x}$ denote the complex conjugate of $x$. Let $\mathbf{x}_1, \mathbf{x}_2, \ldots, \mathbf{x}_n$ denote the training examples. Let $Q$ denote the kernel polynomial so that $K(\mathbf{x}_i, \mathbf{x}_j) = Q(\bar{\mathbf{x}}_i^T \mathbf{x}_j)$. Let $Q(t) = \sum_i q_i t^i$. For simplicity, let us look at the case where the power series and the kernel polynomial are univariate polynomials of a bounded degree $\deg(q)$. We will assume that we have enough samples that Fact 3 hold when averaging over all samples. Let $q_J$ be the coefficient of $T_J$ in the polynomial expansion of $Q(t_1 + \cdots + t_n)$.

**Lemma 9.** *For a univariate polynomial $y = p(x)$, $\bar{\mathbf{y}}^T H^{-1} \mathbf{y} = \sum_j a_j^2/q_j$ asymptotically in the sample size, where $a_j$ are the coefficients of the polynomial $p$. For a multivariate polynomial, $\bar{\mathbf{y}}^T H^{-1} \mathbf{y} = \sum_J a_J^2/q_J$ asymptotically in the sample size. Here, $H^{-1}$ denotes the pseudoinverse of $H$.*

*Proof.* We will begin with the univariate case. Let $\{(x_1, y_1), (x_2, y_2), \ldots, (x_n, y_n)\}$ denote the training examples and their labels. Let $\mathbf{y}$ be the vector of all the labels $\{y_i\}$. Let $d = \max\{\deg(p), \deg(q)\}$ (where we assume that $\deg(q)$ is bounded for simplicity). Now consider the matrix $G$ with $n$ rows and $d$ columns where the $(i,j)$-th entry is $x_i^j$. Note that $\bar{G}^T$ transforms $\mathbf{y}$ from the standard basis to the monomial basis, i.e. the expected value of $(1/n)\bar{G}^T \mathbf{y}$ is $(a_1, \ldots, a_d)$ (by Fact 3). Therefore, $(1/n)\bar{G}^T \mathbf{y} = (a_1, \ldots, a_d)$ asymptotically in the sample size $n$. We claim that $H = GD\bar{G}^T$ where $D$ is the diagonal matrix where $D_{k,k} = q_k$. To verify this, let $G_{(i)}$ denote that $i$-th row of $G$ and observe that the $(i,j)$-th entry $G_{(i)} D \bar{G}_{(j)}^T = \sum_k x_i^k q_k \bar{x}_j^k = q_k (x_i \bar{x}_j)^k = K(x_i, x_j) = H_{i,j}$. Now given the orthonormality of the monomial basis, $(1/n)\bar{G}^T G = I$. Therefore since $H = GD\bar{G}^T$ is the SVD of $H$, $H^{-1} = (1/n^2)GD^{-1}\bar{G}^T$. Hence $\bar{\mathbf{y}}^T H^{-1} \mathbf{y} = ((1/n)G^T \bar{\mathbf{y}})^T D^{-1} ((1/n)\bar{G}^T \mathbf{y}) = \sum_j (1/q_j) a_j^2$.

For the multivariate case, instead of having $d$ columns for $G$, we will have one column for every possible value of $J$ of degree at most $d$. In the diagonal entry $D_{J,J}$ we put $q_J$, where $q_J$ is the coefficient of $T_J$ in the polynomial expansion of $Q(t_1 + \cdots + t_n)$. $\qquad\square$

**Corollary 5.** *For the ReLU activation $q_j = \Omega(1/j)$, and hence $\bar{\mathbf{y}}^T \bar{H}^{-1} \mathbf{y} \geq \Omega(\sum_j j a_j^2)$ asymptotically in the sample size.*

Note that in Theorem 7, the upper bound for the sample complexity was $O(\sum_j j|a_j|)$, hence Theorem 7 is tight up to the distinction between the $\ell_1$ and $\ell_2$ norm (which can differ by at most $\sqrt{\deg(p)}$).

## D  Additional Details for Experiments

### D.1 Setup details

All the experiments are done in TensorFlow, trained with a GPU accelerator. We use the default TensorFlow values for all hyper parameters involved in the training of the neural networks. All the experiment results averaged over 3 runs. The number of training epochs for each experiment and

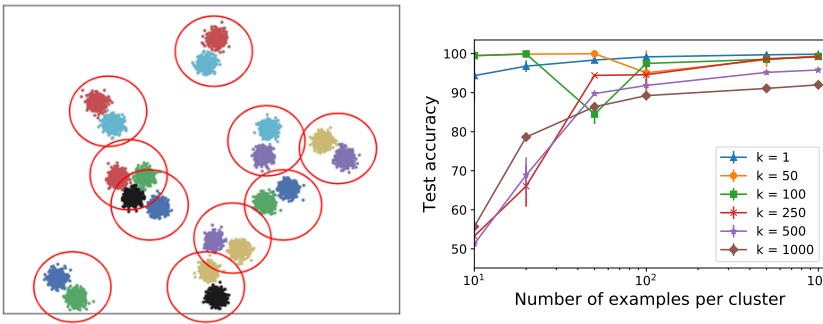

(a) An instance of the problem with multiple clusters, each cluster is indicated by a red circle.

(b) Test accuracy vs. number of points per cluster

Figure 6: Experiment where data is clustered into tasks with a separate linear function for each task. A single neural network does well even when there are multiple clusters.

average runtime (for one run) are summarized in Table 2. For cluster experiments, number of training examples per cluster varies 1000 to 100000, average runtime varies from 2 minutes to 100 minutes. For the decision tree experiments, number of training examples per leaf node varies from 64 to 512, avarage runtime varies from 14 minutes to 42 minutes. For the SQL-style aggregation experiment, the train dataset contains 16384 examples, and test dataset contains 4096 examples, average runtime is 50 minutes. The source for the Penn World Table dataset Feenstra et al. (2015) used in the SQL query experiment is `https://www.rug.nl/ggdc/productivity/pwt/` and it is also available at `https://www.kaggle.com/jboysen/penn-world-table`.

Table 2: Number of epochs and average runtime

| Experiment name | Number of epochs | Average runtime |
| --- | --- | --- |
| Cluster | 100 | 2 - 100 minutes |
| Decision Tree | 200 | 14 - 42 minutes |
| SQL-style aggregation | 6400 | 50 minutes |

### D.2 ADDITIONAL DETAILS FOR LEARNING CLUSTERS OF LINEAR FUNCTIONS

We provide a more detailed setup of the experiment reported in Fig. 3a where the task codes are given by clusters, and there is a separate linear function for every cluster. In this experiment, the data is drawn from $k$ clusters, and from a mixture of two well-separated Gaussians in each cluster. Data points from the two Gaussians within each cluster are assigned two different labels, for $2k$ labels in total. Fig. 6a below shows an instance of this task in two dimensions, the red circles represent the clusters, and there are two classes drawn from well-separated Gaussians from each cluster. In high dimensions, the clusters are very well-separated, and doing a $k$-means clustering to identify the $k$ cluster centers and then learning a simple linear classifier within each cluster gets near perfect classification accuracy. Fig. 6b shows the performance of a single neural network trained on this task (same as Fig. 3a in the main body). We can see that a single neural network still gets good performance with a modest increase in the required number of samples.

