# OpenReview forum: "One Network Fits All? Modular versus Monolithic Task Formulations in Neural Networks"
_ICLR.cc/2021/Conference — ICLR 2021 Poster_

### Official Review · AnonReviewer3 · 2020-10-24
**The paper shows that multiple tasks can be encoded within a neural network that has a monolithic structure. Different ways of encoding the tasks are considered.**

**Rating:** 7
**Confidence:** 3

**Review:**

This paper sets out to show that multiple tasks can be encoded in a neural network that that does not have explicit modular construction for each tasks, which is in contrast with the work of [Bakker and Heskes, JMLR 2003], and [Jocabs, Jordan, Nowlan and Hinton, Neural Computation 1991]. The premise of the paper is that task coding as indicators can be approximated via the derivative of an approximation (erf + Taylor truncation) of the step function. This approximation is analytic and, with individual tasks being analytic, makes the entire multiple task function g(c; x) possible to be approximated by neural networks due the the universal function approximation power of neural networks.

The paper starts from Arora et al. 2019b. However, this seems unnecessary and limiting. The paper can be made more general using from earlier work, such as those since [Hornik 1991].  This is specially so when SGD is not really required by the subsequent developments in the paper.

The paper is lacking much in clarity:
1. The definition of g(c; x) in section 2.2 should be given from the onset and the related work discussion based around that. This will dispels any notion that the paper will dwell on positive/negative transfer/multi-task learning.
2. The paper seems to be a theory paper, but much of the important derivations in the appendix beyond the page limit. For example, Lemma 4 and Theorem 8 are the main support of the paper, and should be in the main part of the paper. Lemma 7 is also important, and it is the closest in the entire paper that answers the question set out in the last sentence of the second para of section 1.
3. The top left box of Figure 1 only makes sense when one reads appendix B1.
4. Overall, the main paper cannot stand on its own without the appendix, which makes it inappropriate for the a conference paper (and its associated review period).

I recommend *reject* based on the clarity and structure of the paper.

COMMENTS AFTER REVISION
=========================
I've looked through the revised paper, and the authors has addressed my comments. Hence, I am happy to recommend an accept.

---

> ### Author Response · Authors · 2020-11-22
> **Response to review**
>
> We thank the reviewer for their comments and for pointing out several very relevant references, which we have now referenced. We would also like to point out that it does not appear that we can begin our analysis from [Hornik’91], since it only concerns the representation power of neural networks. In contrast, we are interested in understanding whether these representations can be learnt from data using efficient algorithms such as SGD.
>
> Regarding the presentation:
>
> We have undertaken several steps to improve the presentation, using the additional 9th page allowed in the rebuttal. We have added more discussion in Section 2.2 and 2.3. As suggested by the reviewer, we have also included Lemma 1 (previously Lemma 4) in the main body now instead of the appendix, since that helps understand some of the basic techniques behind the proofs. We have also added a figure for Section 2.2 and 2.3. We have around half page left on page 9, which we could use to add another figure or proof details if it is helpful.
>
> We are also currently in the process of increasing the number of trials for the experiments and hope to shortly update the experiment results in the revised version.

---

### Official Review · AnonReviewer4 · 2020-10-28
**Good quality and quantity of work but not suitable for conference format.**

**Rating:** 3
**Confidence:** 4

**Review:**


Verdict:

Recommendation to REJECT; Please consider for ICLR Special Journal Issue with modifications.

(Unique situation, please read fully)


##########################################################################
### Summary:

The paper provides insight into the boundaries and feasibilities of a monolithic formulation of multitask learning by
neural networks. The authors show how complex tasks can be modularly formulated thus yielding a joint monolithic
learning possibility. They also show that such modularity can be used to interpret simple algorithms thereby also
leading to their joint learning.

The main contributions of the authors are the following:

- showing that "the two layer neural network can jointly learn the task coding scheme and the task specific
functions without special engineering of the architecture"

- "systematic theoretical investication of the extent of this ability" (ability = single network can successfully
be trained to perform a wide variety of tasks)

- "...primarily interested in the extent to which different tasks may interfere,..." (in a multitask setting)


##########################################################################

### Reasons for recommendation / score:

The paper is composed of a vast amount of very good research work. The research results seem significantly novel and definitely not incremental or based on other similar contemporary works. The rigorous mathematics and the attention to many details is laudable.

However, the content in the paper literally and logically coerces the reader to constantly look at the supplementary material. I could also dare to say that the paper, strictly, without the supplementary material almost seems incomplete or as a compilation of claims, making for a choppy read. This is clearly an effect of the authors trying very hard to squeeze a lot of content into the 8-page limit. This must have taken a lot of efforts and I definitely can see the work that has gone into writing this concisely.

The paper would read much better with all the appendices and supplementary material introduced in the appropriate places, in proper continuum. I could see this paper be re-written almost as a proper tutorial paper in this topic of research. I would in this context, recommend some extra experiments and discussions (commented below also), to make the work more thorough.

For these reasons, I must insist that the venue for submitting this work should be a suitable journal such as JMLR, ML, or specifically be rewritten for the Special Journal Issue @ ICLR 2021. It is not suitable to be accepted as a conference contribution, by the sheer magnitude of work and the style it is presented in.

Another possibility I see, but do not recommend, is that the main motivation of the paper be modified to a solely task solving angle rather than the science of comparison and analysis. Then the theoretical rigour could be reduced and the focus can land on experimental results. This, in my opinion, could be suitable and a definite ACCEPT for ICLR conference.

##########################################################################
### Note about the reviewer:

My area of research is Bayesian non-parametrics applied on to multitask learning. I am not very familiar with the mathematical support provided for the theorems in this work. I cannot promise a critical verification of the correctness of the proofs.

Please also keep this in mind when considering my recommendation above.
##########################################################################
### Pros:

1. The research quality and quantity are exceeding requirements for acceptance!

2. The language is clear and crisp when introducing the research area and placing it in context with its related works. I especially like the delineated "Our Results" section. The authors clarify and discuss the topic with respect to two landmark papers very well.

3. The flow of thought is clear and makes the reader comfortable with the presented paper structure.

4. All the assumptions are made evident and clarified beyond any doubts, there are no hidden assumptions or simplifications. The scope of the focused research is also well clarified.

5. There is thorough mathematical justifications, case studies of monolithic formulations, guarantees on bounds and learnabilities in the supplementary material. (I have tried to not give it attention as it is not a necessary part of the submission)

##########################################################################

### Cons:

1. The title, abstract and some parts of the conclusion suggests a tone of comparison. This makes me expect a more involved discussion about the topic "Modular versus Monolithic Task Formulations". The authors have a lot of insight in this matter, however, when it comes to presentation, they fall short to guide the reader through them.

2. The authors have carefully cherry-picked the theorems and balanced the extent to which they explain them so that it reads with completeness on the whole. I would argue that the details are important and without the proofs and mathematical involvement the scientific reader is forced to question "why?" or "how?" quite often.

3. I would have liked the authors to place themselves better in the research context. I would have liked to know the findings of the authors in the lists of references (in Sections 1.1 and 1.2) with respect to the title of this work.

4. The writing in many middle sections where details are needed, are overtly compressed. This is an effect of trying to squeeze in too much in too little space. The authors do direct the reader to the supplementary material many times.

5. The authors do not answer the grander questions they begin the paper with. They analyse other attributes which are aligned in the same directions as these questions. Eg. Is modular construction better than the monolithic ones? When should we use which construction?

6. The authors talk about multitask learning in the same context as [Caruana 1997]; That is, they reduce the scope of their analysis to inputs of the same size for all tasks or even same inputs to different functions (or tasks) to be learnt. Is this the general case of "any" multitask learning scenario?

7. The experiments themselves do not seem statistically thorough. There should be more than 3 trials to draw conclusions, especially when the experimental setup is based on parameters drawn from a uniform distribution.

8. The authors need to address the few inconsistencies in the graphs they have shown in Fig.2. I definitely would have liked more authors' insight on the observed statistics.

9. In Fig.3 are the numbers significant? They are reported in the third decimal position for the Test R-squared values. Some more explanation is needed.

10. Is the monolithic formulation of multitask learning effectively: joint learning of the switching function and the task function? It would be nice to read some more of the authors' explanation of how and if they are doing something different.


#########################################################################

### Suggested Presentation Changes:

1. There could be more figures explaining the schematics of the networks, explaining the setup etc. Especially Sec.2.3 and Sec.2.4.

2. There could be more figures and clearer captions with simpler explanations. This is where the reader looks first. It would be nice to suggest what to expect from such a graph and then highlight any results.

3. The details in Table 1 can be made more readable. It is unclear where the focus lies.

4. I feel it is important, the authors highlight that the Simple Programming Cosntructs part of their research derives from their novelty in the formulation of modules in terms of mathematical functions!

##########################################################boarman###############
### Suggested Small Corrections:

Generally, try to break down longer sentences into shorter multiple sentences.

- Sec. Abstract: remove or replace word "underlying"

- Sec. Abstract: "... trees over some task-code attributes." Change 'some' --> 'certain'

- Sec. Introduction: "As techniques, such as neural networks, for learning with relatively rick classes have been developed, it is ..."

- Please see if you can move the references to the end of sentences than in the middle. It makes for better readability.

- Sec 1.1: move NTK references to end of the sentence.

- Sec 2.2 theorem 2: for omega(1) far A...A subspaces, if A ...

- Sec 2.2 theorem 2: for omega(1) separated c...c prototypes, if ||c ...

- Fig.3. caption: 1-(Test R-squared). Parentheses helps understand the subtraction from unity.

#########################################################################

Updates:

---

> ### Author Response · Authors · 2020-11-22
> **Response to review**
>
> Thank you for the thoughtful evaluation. We respectfully believe, however, that there is value in presenting work such as this in a conference format, even though there is no way that all of the details of our work can be described under such conditions: it is useful to learn what the main technical claims are and what methods are used to obtain these claims at a high level. We note that this is a relatively common mode of presentation for theoretical work in other machine learning venues such as ICML and NeurIPS.
>
> Regarding questions on the presentation:
>
> We have undertaken several steps to improve the presentation, using the additional 9th page allowed in the rebuttal. We have added more discussion in Section 2.2 and 2.3. We have also included Lemma 1 in the main body now instead of the appendix, since that helps understand some of the basic techniques behind the proofs. As suggested by the reviewer, we have also added a figure for Section 2.2 and 2.3. We have around half page left on page 9, which we could use to add another figure or proof details if it is helpful. We have also made the minor changes and corrections suggested by the reviewer regarding the captions etc.
>
> Regarding some questions about the framework listed in the Cons:
>
> You are correct that our monolithic formulation can be summarized as joint learning of the switching function and task functions. This is the style of formulation that was used, for example, in the work on massively multilingual machine translation (Johnson et al. ‘17, Bapna et al. ‘19). (We stress that we are seeking to capture an approach that is being used in practice, not one that we are proposing!) This formulation is clearly related to, but distinct from, the “multitask learning” formulation originally proposed by Caruana (‘97). Some authors indeed take a more liberal view of what counts as “multitask learning” (e.g., Zhang and Yang survey a wide variety of formulations), and obviously if one chooses to take a sufficiently broad view of what counts, our work would fall under that umbrella as well. On the other hand, it is impossible to nail down a single family of tasks that count as “multitask learning” if one takes such a view, and so comparison with “multitask learning” in such a broad sense becomes a futile exercise.
>
> Most works have found some kind of negative transfer effect when tasks are diverse. For example Johnson et al. (‘17) found a small decrease in BLEU scores for multilingual machine translation as compared to stand-alone training, and Bapna et al. (‘19) found a connection between transfer and interference: namely, when the combined model exhibited transfer from high-resource to low-resource languages, performance for high-resource language suffered. Raffel et al. (‘19) found that combining all of the data sets for a broad variety of tasks in the same way that we are considering was worse than fine-tuning for individual tasks.
>
> These are of course only empirical demonstrations on a specific range of tasks, and thus our main question was how harmful this negative transfer effect can be. Our main findings may be summarized as: at least for sufficiently simple tasks and combining functions, there is indeed a limit to the extent of negative transfer. Although there is indeed often a cost in performance, learning of the monolithic formulation is still possible, and similar in complexity to learning all of the individual tasks.

---

### Official Review · AnonReviewer2 · 2020-10-31
**Nice attempt but very little about the proofs are parsable**

**Rating:** 6
**Confidence:** 3

**Review:**

This paper posits a very interesting question about provable multi-task learning by neural nets. The idea is quite interesting to encode the objectives as task codes and then to write a smooth approximation to the predictor function as a weighted sum of indicators and the try training a net to learn this smoothening. But the problem with the paper is that the presentation of the details are extremely unclear and almost nothing about the proofs can be easily followed! Let me cite a few specific issues,

-  The key theorem here seems to be Theorem 8 on page 18 and it follows from Remark 2 which itself follows from Theorem 7 on page 15 whose proof is claimed to be essentially that of Theorem 6 which in turn follows from equation 9 of Corollary 3 which isn't proven here - and this corollary 3 isn't the same as the corollary 6.2 of https://arxiv.org/pdf/1901.08584.pdf because this allows for odd powers and maybe more importantly the definition of H in this corollary 3 has no analogue in 1901.08584 and hence its mysterious to me as to where does this come from! What has the H in this corollary 3 got to do with the H^\infity of 1901.08584 which is the NTK kernel. Its far from obvious that the proof there will go through here again. So as it stands the information given in the paper makes it difficult to trust that the result is correct. I would strongly suggest resubmitting to a future venue a self-contained paper with all the proofs. This writing here is looking too messed up!

- The argument between equation 22 and 23 also make similar claims about analogy to proofs in the paper https://arxiv.org/pdf/1901.08584.pdf but here its not even clear as to which proof in there is being referred to!

- Around equation 4 a claim is being made that https://arxiv.org/pdf/1901.08584.pdf has training only on the second layer for randomly initialized bottom layer. This doesnt seem to be true! Can the authors point to this assumption anywhere in that paper? 1901.08584 is not a NNGP result!

- The proof of Theorem 14 claims that SGD can be implemented as a SQ algorithm. Can you reference this claim? If not, it would be best to put in the proof here since this isnt obvious at all.

---

> ### Author Response · Authors · 2020-11-22
> **Response to review**
>
> We apologize for the confusion presentation of the proofs, and have uploaded a new version (particularly in Appendix A). We address some of the specific issues below.
>
> With regards to the flow from Corollary 3 to Theorems 6, 7, and 8: we have rewritten the proofs to make more clear which are our new contributions, and how they follow from previous work. To summarize, we show the following:
>
> 1. Theorem 3.3 from https://arxiv.org/pdf/1901.08584.pdf, proven for single-hidden layer neural networks trained with full batch GD with only the hidden layers trained (NTK learning), also applies to networks with only the last layer trained (NNGP kernel learning) (our new *Theorem 6*). The main utility of this theorem is to compare theoretically the performance of kernel methods with the performance of neural network methods.
> 2. We then prove *Corollary 3*, which gives learning bounds for functions expressible as power series. This is a more general version of Corollary 6.2 from https://arxiv.org/pdf/1901.08584.pdf, in particular extended to apply to NNGP kernels and to all powers of monomials (instead of merely even ones).
> 3. We then use *Corollary 3* to prove *Theorem 7* (previously *Theorem 6*) to get learning bounds on analytic functions.
> 4. We then develop Lemma 3 explicitly, which allows us to prove *Theorem 8* (previously *Theorem 7*), which extends the learning bounds to multivariate functions.
>
> This then provides the theoretical basis for the rest of the proofs in the paper. We believe that this allows us to present our results while not explicitly reproducing large amounts of text from previous works.
>
> The proof of Lemma 3 (equations 22 and 23) are analogous to Appendix E (“Proof of 6.1”) of the cited work. We have made that explicitly clear in the text as well.
>
> SGD as a SQ algorithm: We have added a citation to make this clear: “On the Complexity of Learning Neural Networks” (Song-Vempala-Wilmes-Xie’17) mentions that SQ “includes all known variants of SGD with any loss function”.

---

### Official Review · AnonReviewer1 · 2020-11-06
**Perspective is Interesting but Writing is Confusing**

**Rating:** 5
**Confidence:** 2

**Review:**

This paper takes an interesting theoretical dive on the learnability of multiple tasks as one task, while the tasks are constructed with special structures of the cluster, decision tree, or simple program. This paper provides sample complexities analysis, showing that wide two-layer neural networks with standard activation functions and SGD optimization is able to capture the data regularity between input and output, modulated by the three kinds of task code considered. Experiment results show that the networks are flexible enough to fit complex data generated in this way.

Pros:

* This paper is quite novel in many aspects, including modularity v.s. monolithic, constructing task codes by SQL-style aggregation queries, "inverse counterpart" of multitask learning, connections to cognitive science.

* This paper provides impressive sample complexity analysis, with both informal and formal versions.

Cons:

* The clarity of writing is low. There are many places unclear, in particular, the "task distinctiveness". If we view it as one complex task, with the input-output data relationship modulated by one of the three task codes, then the questions of "learnability" by neural networks in handling this case are well addressed by the proposed theoretical analysis and I can see the empirical study is supportive of this claim. But if we interpret them as multiple tasks, how do we know the network "is capable of simultaneously learning multiple tasks" instead of just one task?

* Possible gaps between claims and experiment

Related to the question above, it seems very attractive to me that "the two-layer network can jointly learn the task coding scheme and the task-specific functions without special engineering of the architecture", but how to justify "task-specific" functions/task code has been successfully learned?

It seems in Section 2, it is claimed that "we wish to learn a function g such that g(c^i; x) = f_i(x) from the examples of the form (c^(i); x, f_i(x))". Is c known? Is the learned network being able to take task code as input? But it seems the experiment shows f is learnable not g.

Questions:

* How is the distinctness of tasks defined? Although the examples shown in Figure 1 are quite heterogeneous, examples considered such as  "learning binary classification for well-separated data" resembles mixture-of-experts classification, which can be viewed as one task to learn directly.

* How the task coding is determined by a balanced decision tree? The setup in Section 2.3 is quite confusing. Moreover, is the construction of the g function different from the cluster setting? How does the complexity of task coding schemes influence the way the g function is constructed? This paper would benefit from explaining the different complexities added to the learning by different task coding schemes, and how do they make multiple tasks potentially conflict/distinct with each other.

Overall, I think the monolithic task formulation is novel and the problems the authors would like to address are fundamentally important. I am not from the theoretical machine learning side so I am not sure about the technical contribution of the theorems. The experiments look only partially supportive of the main claims. It is very likely for me to miss something important. The lacking of clarity is a big downside.

---

> ### Author Response · Authors · 2020-11-22
> **Response to Reviewer 1**
>
> Sorry for the confusion. The task code is given together with the feature values and label for each example, essentially as an additional set of features in the example. So, the network is given both the task code c^i and x as inputs, and trained to fit the label f_i(x). You could take the view that this is a more complex single classification task. But, the individual tasks are given by the subsets of the data with common (or close) task code features, which are obtained by construction, again capturing what is done in practice.
>
> In both the clustered data example and the decision tree example, the labels are given by first computing a function of the task code features to identify a task i, and then computing the task function f_i on the (task-specific) features, x. The complexity of these task code functions is indeed different in the two cases, impacting the overall complexity of the monolithic classifier g.

---

### Decision · Program_Chairs · 2021-01-07
**Final Decision**

**Decision:**

Accept (Poster)

**Comment:**

This paper shows how multiple tasks can be encoded in a single neural network without the need for explicit modular construction for each task. The idea is very interesting and the research work presented is of high quality.

All the reviewers underline their interest in the presented work. However, there is a deviation in the reviewers' score with half voting
for acceptance and the other half for rejection. The main concern of the fellow reviewers with the below acceptance threshold score was the difficulty in grapsing the theory of the research presented due to the lack of important content from the main manuscript due to space limitations. The authors have an extended supplementary material that covers the whole magnitude of their work.

I understand the reviewers' concern on how such a dense presentation does not do justice and harms the presented effort itself. However, given the edits the authors added to address the issue rasied and the interest and potential of this work - acknowledged by all the reviewers and myself I recommend acceptance. This is a work of a quality I would like to keep seeing in ICLR.